# The impact of learning on perceptual decisions and its implication for speed-accuracy tradeoffs

André G. Mendonça[1,4], Jan Drugowitsch [2,4], M. Inês Vicente[1], Eric E. J. DeWitt [1], Alexandre Pouget [3,5] & Zachary F. Mainen[1,5 ✉]

In standard models of perceptual decision-making, noisy sensory evidence is considered to be the primary source of choice errors and the accumulation of evidence needed to overcome this noise gives rise to speed-accuracy tradeoffs. Here, we investigated how the history of recent choices and their outcomes interact with these processes using a combination of theory and experiment. We found that the speed and accuracy of performance of rats on olfactory decision tasks could be best explained by a Bayesian model that combines reinforcement-based learning with accumulation of uncertain sensory evidence. This model predicted the specific pattern of trial history effects that were found in the data. The results suggest that learning is a critical factor contributing to speed-accuracy tradeoffs in decision-making, and that task history effects are not simply biases but rather the signatures of an optimal learning strategy.

[1] Champalimaud Research, Champalimaud Centre for the Unknown, Lisbon, Portugal. [2] Neurobiology Department, Harvard Medical School, Boston, MA, USA. [3] University of Geneva, Geneva, Switzerland. [4]These authors contributed equally: André G. Mendonça, Jan Drugowitsch. [5]These authors jointly supervised: Alexandre Pouget, Zachary F. Mainen. ✉email: zmainen@neuro.fchampalimaud.org

Evidence accumulation is an important core component of perceptual decision-making that mitigates the effects of environmental uncertainty by combining information through time[1–8]. Theoretical models based on a random diffusion-to-bound (Drift-diffusion models—DDMs) have been successful in modeling critical aspects of psychophysical decision tasks, capturing the dependence of accuracy (psychometric) and reaction time (chronometric) functions. These models have been tested both by searching for neural activity corresponding to model variables[1,9–13], and the exploration of more sophisticated task designs and modeling[6,14].

One widely observed but not well-understood phenomenon is that different kinds of decisions appear to benefit from accumulation of evidence over different time scales. For example, monkeys performing integration of random dot motion[1] and rats performing a click train discrimination task[6] can integrate evidence for over one second. But rats performing an odor mixture categorization task fail to benefit from odor sampling beyond 200–300 ms[14,15]. A possible explanation is that neural integration mechanisms are specific to a given species and sensory modality. However, even animals performing apparently similar odor-based decision tasks can show very different integration time windows[16,17]. Changes in speed-accuracy tradeoff (SAT)[2,11,18], which could change the height of the decision bound, have been proposed as a possible explanation for differences seen across similar studies. However, manipulation of motivational parameters failed to increase the integration window in odor categorization, suggesting that other factors must limit decision accuracy[14].

In DDMs, the chief source of uncertainty is stochasticity in incoming sensory evidence, modeled as Gaussian white noise around the true mean evidence rate[19,20]. It is this rapidly fluctuating noise that accounts for the benefits of temporal integration. The nature and implications of other sources of variability have also been considered[6,8,19–22], including variability in starting position[21], non-accumulation time[20] and threshold[19]. A potentially important source of variability is trial-by-trial fluctuations in the mean rate of evidence accumulation. Such fluctuations would correspond to uncertainty in the mapping of sensory data onto evidence for a particular choice[9,23]. This mapping could be implemented as the strength of weights between sensory representations into action values[9]. A combination of weights would then represent a classification boundary between sensory stimuli[24]. Weight fluctuations would introduce errors that, unlike rapid fluctuations, could not be mitigated by temporal integration and would therefore curtail its benefits[14,25]. Such "category boundary" variability (not to be confused with the stopping "bound" in accumulation models) might affect differently particular decision tasks, being particularly important when the stimulus-to-action map must be learned de novo[14,25].

The effects of reward-history on choices in perceptual tasks, although commonly observed[26–29], have been considered suboptimal biases because each trial is in fact independent of the preceding trials. Here, we hypothesized that such biases are instead signatures of an optimal learning strategy that is adapted to natural dynamic environments[30]. Intuitively, an optimal learning agent must always use both priors (history of stimuli, choices and rewards) and current sensory information in proportion to their confidence[31,32]. Under this view, an optimal choice policy and learning algorithm that uses accumulation of evidence and reward statistics to infer choices and update its stimulus-choice mapping—a Bayesian drift-diffusion model—can be derived[32]. Here, to test this model, we compared performance of rats in two odor-guided decision tasks: (1) an odor identification task in which the difficulty was increased by lowering stimulus concentration and (2) an odor mixture categorization task[15], in which the difficulty was increased by making the stimuli closer to a category boundary. We hypothesized that performance in the second task would be dominated by uncertainty in the stimulus-choice mapping and therefore benefit less from sensory integration. Indeed, we observed that the change in reaction times over a given range of accuracy was much smaller in the mixture categorization task. We found that standard diffusion-to-bound models could fit performance on either task alone, but not simultaneously. However, the optimal Bayesian-DDM model could fit both tasks simultaneously and out-performed simpler models with and without alternative learning rules. Critically, the introduction of learning predicted a history-dependence of trial-by-trial choice biases whose specific pattern was indeed observed in the data. These findings suggest that "errors" in many psychophysical tasks are not due to stochastic noise, but rather to suboptimal choices driven by optimal learning algorithms that are being tested outside the conditions in which they evolved[33].

## Results

**Different speed-accuracy tradeoffs in two different olfactory decision tasks.** We trained Long Evans rats on two different versions of a two-alternative choice (2AC) olfactory reaction time task. We refer to these as two "tasks", but they were identical in all aspects except for the nature of the presented stimulus (Fig. 1). In the first task, "odor identification", a single pure odor was presented in any given trial. We manipulated difficulty by diluting odors over a range of 3 log steps (1000-fold, liquid dilution) (Fig. 2a). Thus, absolute concentration determined the difficulty. In the second task, "odor categorization", mixtures of two pure odors were presented with a fixed total concentration but at four different ratios[15] (Fig. 2b). The distance of the stimulus to the category boundary (50/50, iso-concentration line), determined the difficulty, with lower contrasts corresponding to more difficult

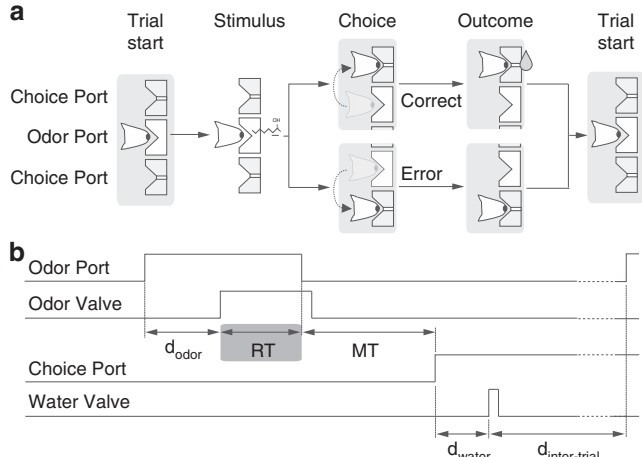

**Fig. 1 Two-alternative odor choice task. a** Rats were trained in a behavioral box to signal a choice between left and right port after sampling a central odor port. The sequence of events is illustrated using a schematic of the ports and the position of the snout of the rats. **b** Illustration of the timing of events in a typical trial. Nose port photodiode and valve command signals are shown (thick lines). A trial is initialized after a rat pokes into a central Odor Port. After a randomized delay $d_{odor}$ a pure odor or a mixture of odors is presented, dependent of the task at hand. The rat can sample freely and respond by moving into a choice port in order to get a water reward. Each of these ports is associated to one of two odors—odor A ($(R)$-$(-)$-2-octanol) and odor B ($(S)$-$(+)$-2-octanol). Highlighted by the gray box, reaction time (RT) is the amount of time the rats spend in the central Odor Port after odor valve is on (i.e. discounting $d_{odor}$). See Methods for more details.

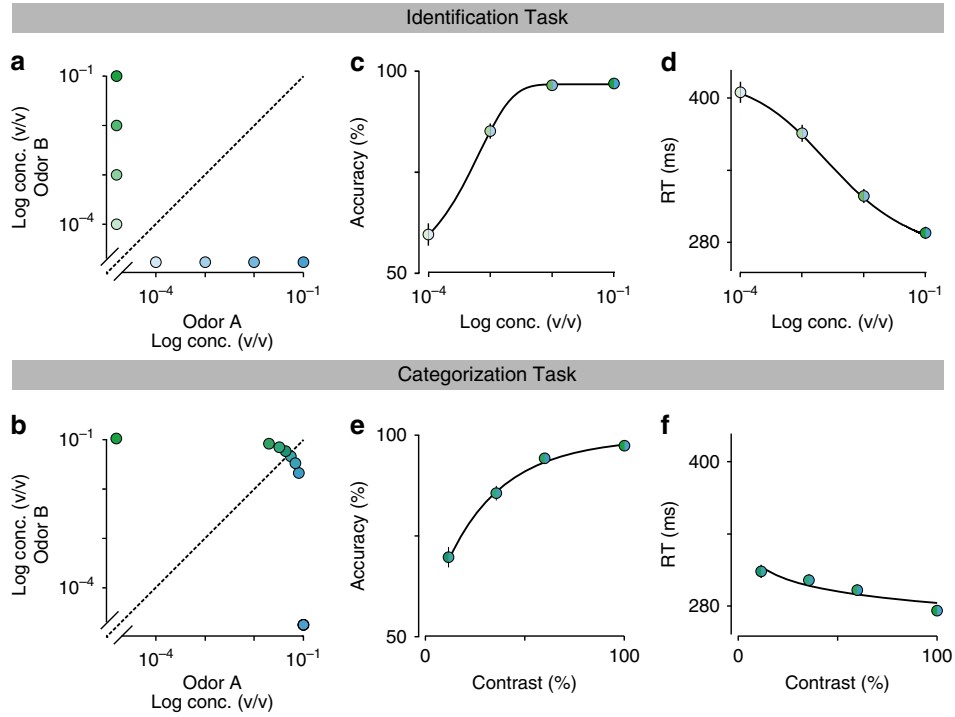

**Fig. 2 Comparison between odor identification and mixture categorization tasks. a**, **b** Stimulus design. In the odor identification task, the odorants were presented independently at concentrations ranging from $10^{-1}$ to $10^{-4}$ (v/v) and sides rewarded accordingly (**a**). For the mixture categorization task, the two odorants were mixed in different ratios presented at a fixed total concentration of $10^{-1}$, and rats were rewarded according to the majority component (**b**). Each dot represents one of the 8 stimuli presented for each task. **c**, **d** Mean accuracy (**c**) and mean reaction time (**d**) for the identification task plotted as a function of odor concentration. **e**, **f** Mean accuracy (**e**) and mean reaction time (**f**) for the categorization task plotted as a function of mixture contrast (i.e. the absolute percent difference between the two odors). Error bars are mean ± SEM over trials and rats. Colors in dots are presented as to help parse between stimulus space and psych- and chronometric curves. Solid lines depict the obtained fits for the predicted curves of a DDM, an exponential curve for performance and a hyperbolic tangent for RTs, as described in ref. [2].

trials. E.g., 56/44 and 44/56 stimuli (12% contrast) were more difficult than 80/20 and 20/80 (60% contrast). Note that the easiest stimuli ($10^{-1}$ dilution and 100% contrast) were identical between the two tasks. In a given session, the eight stimuli from one of the two tasks were presented in randomly interleaved order. To ensure that any differences in performance were due to the manipulated stimulus parameters, all comparisons were done using the same rats performing the two tasks on different days with all other task variables held constant (Supplementary Fig. 1). We quantified performance using accuracy (fraction of correct trials) and odor sampling duration, a measure for reaction time (RT)[14,15] (Fig. 1, Supplementary Fig. 2). We observed that rats performing the two tasks showed marked differences in how much RTs increased as task difficulty increased (Fig. 2c–f). For the identification task, RTs increased substantially (112 ± 3 ms; mean ± SEM, n = 4 rats; F(3,31) = 44.04, P < $10^{-7}$; Fig. 2d), whereas for the categorization task the change was much smaller (31 ± 3 ms; F(3,31) = 2.61, P = 0.09, ANOVA) (Fig. 2f), despite the fact that the accuracy range was similar.

To control for the possibility that a smaller performance range for the categorization task accounted for differences in SAT, we re-ran this task with two sets of stimuli with harder, lower contrast stimuli. This yielded a range of accuracies as broad as those in the identification task yet still only resulted in 41 ± 24 and 50 ± 19 ms changes in RT (Supplementary Fig. 3). Therefore, the difference observed in SAT for odor identification vs. mixture categorization was not due to differences in the range of task difficulties.

**Construction of a diffusion-to-bound model for olfactory decisions**. In order to explore which variables might be

constraining the rats' performance, we fit the data using DDMs (Fig. 3a). In a 2-AFC task with free response time, trading off the cost of accumulating evidence with reward rate becomes paramount. With adequately tuned decision thresholds, DDMs are known to implement the optimal tradeoff strategy across a wide range of tasks, including those used here[34–36]. We implemented a DDM composed of sensory, integration and decision layers. The sensory layer implements a transformation of odor concentrations into momentary evidence. Perceptual intensity in olfaction[37,38], as in other modalities[2,6] can be well-described using a power law. We therefore defined the mean strength of sensory evidence $\mu$ for each odor using a power law of the odor concentrations,

$$\mu_i(c_i) = kc_i^{\beta} \tag{1}$$

where $k$ and $\beta$ are free parameters[2]. We constrained $k$ and $\beta$ to be identical between the two odors (stereoisomers with identical vapor pressures and similar intensities[15,39–41]). Evidence at each time step is drawn from a normal distribution $m_i(t) : N(\mu_i, \sigma)$, where $\sigma = 1$ is the standard deviation of the variability corrupting the true rate, $\mu_i$. The integration layer, which also consists of two units, integrates the noisy evidence over time independently for each odor. The last step of the model consists of a unit that takes the difference between the integrated inputs. If this difference exceeds a given bound, $\theta$ or $-\theta$, the model stops and makes a choice according to the hit bound: left for $\theta$, right for $-\theta$. Finally, we allowed for a time-dependent decrease in bound height ("collapsing"), $\tau$, mimicking an urgency signal[35,42] (Methods).

**Diffusion-to-bound model fails to fit both tasks simultaneously**. To explain our behavioral data with the standard DDM,

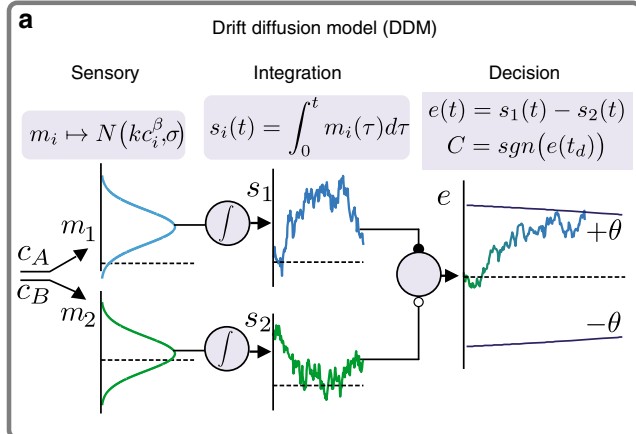

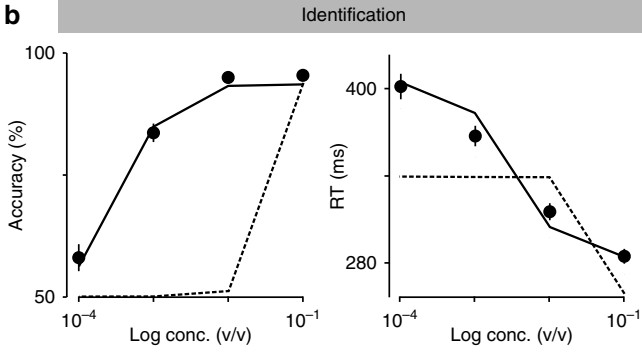

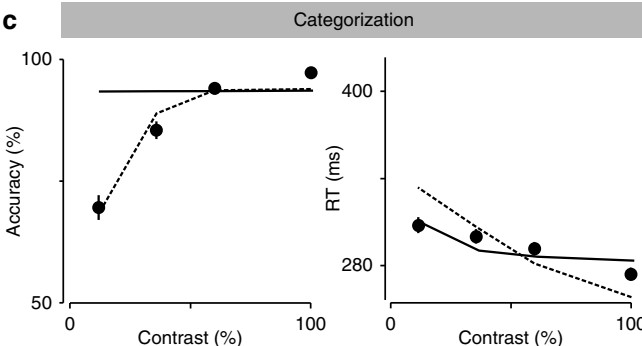

**Fig. 3 Failure to simultaneously fit performance on identification and categorization tasks with Drift-diffusion model. a** Drift-diffusion model (DDM). The model consists of three layers—sensory, integration and decision layer. At the sensory layer, concentrations are transformed into rates that are contaminated with Gaussian noise. These rates are then integrated over time (integration layer) and combined. Note that the choice of weights (−1 and 1) for the decision layer allows it to effectively be a DDM with collapsing bounds. This model presents 7 parameters (Methods). **b** Fitting results for accuracy and reaction time in identification task. Black solid line represents the model fit for this data, and dashed lines the prediction from the categorization data fit. **c** Fitting results for accuracy and reaction time in categorization task. Solid black lines depict the prediction for this data from the model fitted to identification, and dashed lines the DDM fit for this data. Error bars are mean ± SEM over trials and rats.

we developed a series of different fitting procedures. All involved maximizing a log-likelihood function for a data set of 22,208 (identification), 19,947 (categorization) or 42,155 (both) trials using simulations over 100,000 trials (Methods). The overall quality of each fit is shown in Supplementary Fig. 4. The first procedure was to test whether we could predict the behavioral data

of the categorization task using the fitted parameters from the identification task. The model captured both accuracy and RTs in the identification task (Fig. 3b, solid lines). However, when the same model was run on the categorization task, the model correctly predicted the range of RT's in the data, but strongly overestimated the animals' accuracy at low contrasts (Fig. 4c, solid lines). Therefore, as a second procedure, we attempted to fit the model to the categorization task and predict the identification task. This was also unsuccessful: the model fit the categorization data well (Fig. 3c, dashed lines) but failed to capture either accuracy or RTs in the identification task (Fig. 3b, dashed lines). A third procedure, simultaneous fitting both data sets, also failed in describing both tasks successfully (Supplementary Fig. 5). Thus it was not possible to accurately fit the standard DDM to both tasks using a single set of parameters. Accurate fits to both tasks were only possible if we allowed parameters to be fit independently.

**Differences in SAT are not due to context dependent strategies.** Motivational variables can modulate performance and reaction time in perceptual tasks. For example, variables like reward rate[35] or emphasis for accuracy vs. speed[2,11] can have an effect on observed SATs, by modulating decision criteria. Because the two tasks were run in separate sessions, we considered the possibility that rats changed these criteria between sessions. To address this, we devised a "mixture identification" task in which we interleaved the full set of stimuli from the two tasks as well as intermediate mixtures (Fig. 4a). On any given session, 8 randomly chosen stimuli out of the 32 possible were presented. Consistent with the previous observations, RTs in this joint task were significantly affected by concentration but not by mixture contrast (Fig. 4b, c; two-way ANOVA ($F(3,48) = 8.69$, $P < 10^{-3}$ vs. 0.94, $P = 0.42$)). There was no significant interaction between concentration and contrast ($F(9,48) = 0.28$, $P > 0.9$). Each individual rat showed a significant effect of odorant concentration (ANOVA for each rat: $F_1(3,15) = 78.66$, $P_1 < 10^{-6}$; $F_2(3,15) = 14.66$, $P_2 < 10^{-3}$; $F_3(3,15) = 204.91$, $P_3 < 10^{-7}$; $F_4(3,15) = 27.86$, $P_4 < 10^{-4}$), whereas only two showed a significant effect of mixture contrast ($F_1(3,15) = 1.14$, $P_1 = 0.39$; $F_2(3,15) = 0.52$, $P_2 = 0.67$; $F_3(3,15) = 9.6$, $P_3 < 0.01$; $F_4(3,15) = 6.47$, $P_4 < 0.05$). These results indicate that the differences in the relation between accuracy and RT in the previous data set are not due to changes in decision criteria across sessions. As expected from the failure of standard model to fit the previous data, the standard DDM model could not explain these data either (Supplementary Fig. 6).

**DDM with stimulus-dependent Bayesian learning fits performance across both tasks.** Until now, we have been considering a standard DDM that assumes all behavioral uncertainty comes from rapid variability in incoming sensory evidence. However, it is well known that subjects' choices are sensitive to the recent history of rewards[28,43–45], and that reward expectation can influence performance and RTs[14,46,47]. One possible explanation for the overestimate of accuracy in the categorization task is therefore that choices and trial outcomes produce on-going fluctuations in the animals' mapping from odors to choices through a process resembling reinforcement learning. Such fluctuations would produce uncertainty in classification of stimuli near the category boundary that could not be rescued by integration during a trial[14,25].

To develop this idea, we asked how optimal subjects ought to use trial history (stimuli, choices, and rewards) to update their "belief" about the category boundary under the assumption that it is volatile (i.e. that the true mapping from stimuli to correct choices varies stochastically across trials). Although the full Bayesian optimal strategy is intractable, we were able derive a

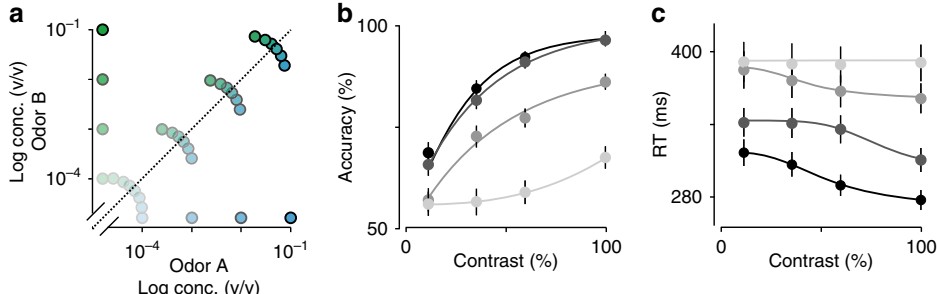

**Fig. 4 Odor mixture identification task. a** Stimulus design. Two odorants ($S$-($+$)-2-octanol and $R$-($-$)-2-octanol) were presented at different concentrations and in different ratios as indicated by dot positions. In each session, four different mixture pairs (i.e. a mixture of specific ratio and concentration and its complementary ratio) were pseudo-randomly selected from the total set of 16 mixture pairs and presented in an interleaved fashion. **b**, **c** Mean accuracy (**b**) and mean of reaction times (**c**) plotted as a function of mixture contrast. Each point represents a single mixture ratio. Error bars are mean ± SEM over trials and rats. Solid lines depict the obtained fits for the predicted curves of a DDM for each set of mixtures at a particular concentration, family functions as described in ref. [2]. Colors represent the total concentration of the mixture, with black indicating a $10^{-1}$ mixture and lightest gray $10^{-4}$ mixtures.

near-optimal strategy that yields behavioral performance indistinguishable from optimal[32] (Methods). This resulted in a DDM with stimulus-dependent Bayesian learning, which we refer to as "Bayes-DDM" (Fig. 5a). The Bayes-DDM is the same as the standard DDM but augmented with weights that transform the stimulus input into evidence:

$$e_i = w_i s_i(t),\qquad(2)$$

which is then combined with bias $b$ to form a net evidence

$$e(t) = w_1 s_1(t) + w_2 s_2(t) + b.\qquad(3)$$

In this equation, the weights $w_i$ and the bias $b$ define, respectively, the slope and offset of the category boundary.

After each trial, we updated the stimulus weights $w_i$ using a tractable approximation to the Bayes-optimal learning rule,

$$\Delta \boldsymbol{w} = \alpha_w(\boldsymbol{s}, t)\Sigma_w\boldsymbol{s},\qquad(4)$$

where $\Sigma_w$ is the weight covariance matrix (also learned; Methods), that quantifies the current weight uncertainty, and $\alpha_w(\boldsymbol{s}, t)$ the learning rate. This learning rule introduces three new parameters that describe the learner's assumptions about how the weights change and influence the learning rate $\alpha$ (Methods).

We fitted the Bayes-DDM to the data by maximizing the log-likelihood of both olfactory decision tasks simultaneously[2]. In the absence of a closed-form analytical solution[19], we generated mean RTs, choices and trial-to-trial choice biases by numerically simulating a sequence of 100,000 trials for each combination of tested parameters (Methods). In contrast to the standard DDM, the Bayes-DDM produced a very good simultaneous fit of both tasks (Fig. 5b, c). As a further test, we also assessed whether the model could fit the behavioral results for the merged (interleaved) task (Fig. 4). To do so, we fitted the model to the 32 stimuli from the interleaved condition. We found that the model indeed provided a good qualitative match to this data set as well (Fig. 5d). Therefore, by making the additional assumption that subjects assume a volatile category boundary and make trial by trial adjustments accordingly, we were able to arrive at a model that captured our entire data set.

**Bayes-DDM successfully predicts trial-by-trial conditional changes in choice bias.** The Bayes-DDM model can be considered as a hypothesis concerning the form of trial-to-trial biases that we expect to be sufficient to explain the data. Crucially, the specific predictions of this model can be tested against behavioral variables that were not directly fit. That is, we can check whether

the form of the trial-to-trial biases in the experimental data is in fact compatible with the form and magnitude of the learning we introduced.

We observed quite large effects of trial history. Figure 6 shows the average psychometric choice functions (Fig. 6a, b, dashed lines) and psychometric choice functions conditioned on the previous odor stimulus (Fig. 6a, b, solid lines, with different stimulus difficulties separated by quadrants, as indicated). Note that only cases in which the previous trial was rewarded are included. To quantify the impact of a previous trial, we calculated the difference in the average choice bias conditional upon the trial being correct and a given stimulus being delivered relative to the overall average choice bias ($\Delta C_B(x)$; Methods). Note that $\Delta C_B(x)$ is a measure of the amount of learning induced by a past trial, as measured the fractional change in choice probability, with $\Delta C_B(x) > 0$ indicating a greater likelihood of repeating a choice in the same direction as the prior trial, $\Delta C_B(x) < 0$ a choice in the opposite direction. Because $\Delta C_B(x)$ was symmetric for left/right stimuli, we plot $\Delta C_B(x)$ collapsed over stimuli of equal difficulty (Fig. 6c, d; uncollapsed data plotted in Supplementary Fig. 7; individual rats shown in Supplementary Fig. 8).

These analyses showed that rats have a tendency to repeat a choice in the same direction that was rewarded in the previous trial ("win-stay"), but the stimulus-dependent analysis revealed a qualitative difference between the two tasks with respect to how past stimuli impacted choice bias. For the identification task, the influence of the previous trial was largely stimulus-independent (Fig. 6c, one-way ANOVA, $F(3,12) = 2.0$, $P = 0.17$). For the categorization task (Fig. 6d), in contrast, that influence showed a graded dependence on the stimulus, being larger for a difficult previous choice than for an easier one ($F(3,12) = 25.4$, $P < 10^{-5}$). We also conducted this analysis for incorrect trials but, due to the small numbers of trials, the data were too variable to draw any firm conclusions (Supplementary Figs. 9, 10).

Remarkably, for both tasks the predictions of Bayes-DDM closely matched the data. For the categorization task, as expected, the model captured the strong dependence of $\Delta C_B(x)$ on stimulus difficulty (Fig. 6d). For the identification task, the model was able to capture the relative lack of stimulus dependence of $\Delta C_B(x)$ (Fig. 6c). These results can be understood by considering that the Bayes-DDM depends on both the accumulated inputs $\boldsymbol{s}$ and decision time $t$, reflecting a form of decision confidence[32] (Methods). In tasks like ours, with a varying difficulty, harder trials are associated with later choices and come with a lower decision confidence[48]. On correct easy trials, learning is smaller when the animal's confidence is high. This makes sense: if the

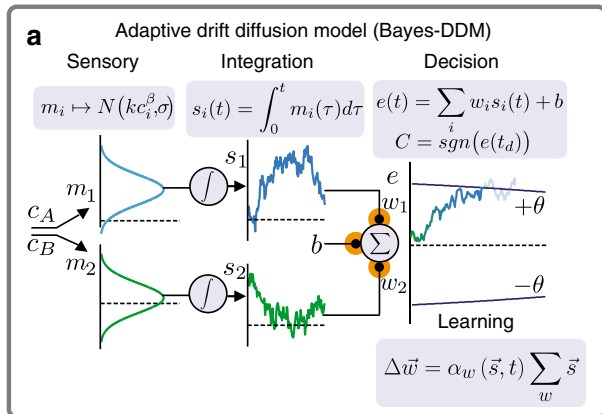

**Fig. 5 Bayesian-DDM with bias and stimulus learning explains identification and categorization task simultaneously. a** DDM is even further expanded with the addition of changing stimulus weights, $w_1$ and $w_2$, and trial-by-trial reward-dependent bias $b$. These weights are then combined with the integrated momentary evidences ($s_1, s_2$) plus the offset set by the bias $b$. After each trial the model updates stimulus weights according to the obtained outcome through a Bayesian learning rule. This model has 10 parameters (Methods). **b**, **c** Choice accuracy (fraction of correct trials) and odor sampling duration in identification task (**b**) and categorization task (**c**). Solid black line represents model fitted to both tasks (see Methods for more details). **d** Choice accuracy and odor sampling duration for Interleaved condition. Solid lines represent the obtained fits for Bayes-DDM to this particular data, going from lowest total concentration (lightest gray) to highest (black). Error bars are mean ± SEM over trials and rats.

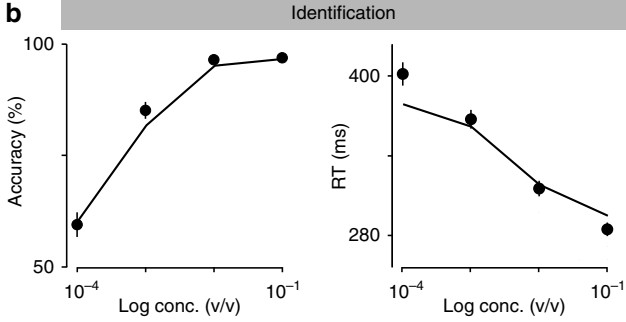

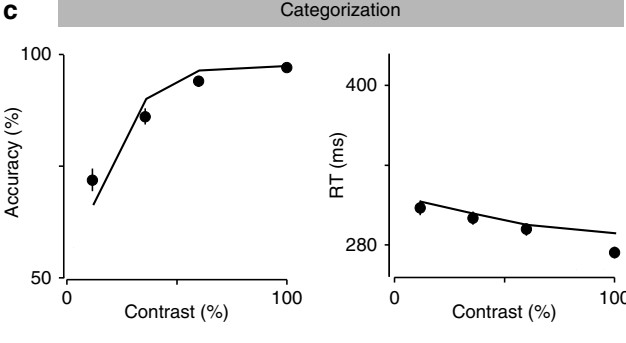

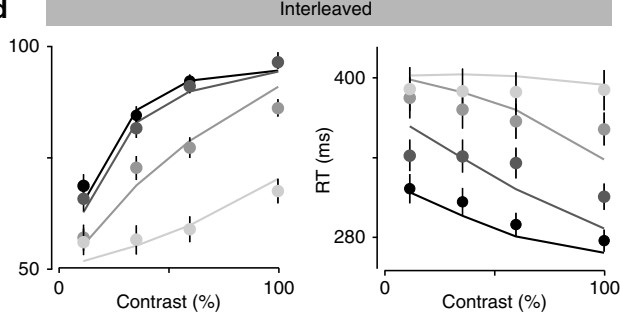

animal is correct and highly confident, there is little reason to adjust the weights. The relative lack of stimulus dependence of $\Delta C_B(x)$ in the identification task is explained by a low signal-to-noise ratio for difficult trials, implying that the sensory component of Eq. 4 will be low. Thus, there is a larger contribution of the stimulus-independent term (the bias—$b$) in updating choice bias (Methods). Although the dependence of decision confidence on decision time suggests the possibility of a dependence of $\Delta C_B(x)$ on RT, we found that bias learning washes out the confidence–RT relationship, such that neither data nor the Bayes-DDM model feature a strong modulation of learning by RT (Supplementary Fig. 11).

**Comparison with other learning rules**. The optimal Bayes-DDM learning rule takes a complex form involving multiple terms whose respective roles are not immediately clear. In order to gain some insight into why this rule captures the animals' behavior, and whether confidence has a role, we fitted several variations of our model.

We first fitted a model without learning but in which the weights are drawn on every trial from a multivariate Gaussian distribution whose mean is set to the optimal weights ($1/\sqrt{2}$, $-1/\sqrt{2}$ and $0$) and whose variance is a free parameter. Interestingly, this model could fit the psychometric and chronometric curves in both tasks. However, the model failed to show sequential effects since the weights are redrawn independently on every trial (Supplementary Fig. 12). Model comparison confirmed that this model performs considerably worse than Bayes-DDM (Fig. 7).

DDMs weight the difference between the two evidences. To control for a possible alternative integration process, we implemented two different versions of an LCA model[49] (Methods) in which absolute evidences for each side are integrated over time, and the two integrators inhibit each other. These models did a good job of explaining both psych- and chronometric curves. but failed to replicate the changes in choice bias seen in the data (Supplementary Figs. 13, 14). This also suggests that a learning process must be at play.

Next, we tried a model with a limited form of learning in which the optimal learning rule is applied only to the bias while the sensory weights are set to their optimal values. It has been argued that sequential effects can be captured by variations in the bias[28]. This model had a BIC score comparable to the optimal model (Fig. 7) and captured the flat profile of the identification task, thus suggesting that sequential effects in this task are due to bias fluctuations. However, this model failed to account for the profile of sequential effects in the categorization task (Supplementary Fig. 15). Although the data show sequential effects inversely proportional to the difficulty of the previous trial, this model predicted a flat profile (Supplementary Fig. 15g).

Conversely, we fitted a model in which the sensory weights, but not the bias, are adjusted on every trial according to the optimal learning rule. This model fit the psychometric and chronometric curves reasonably well (Supplementary Fig. 16). However, in contrast to the previous model, this one captured the sequential effects in the categorization task but not in the identification task (Supplementary Fig. 16d). Moreover, the BIC score for this model was far worse than Bayes-DDM. In addition, we fitted a model

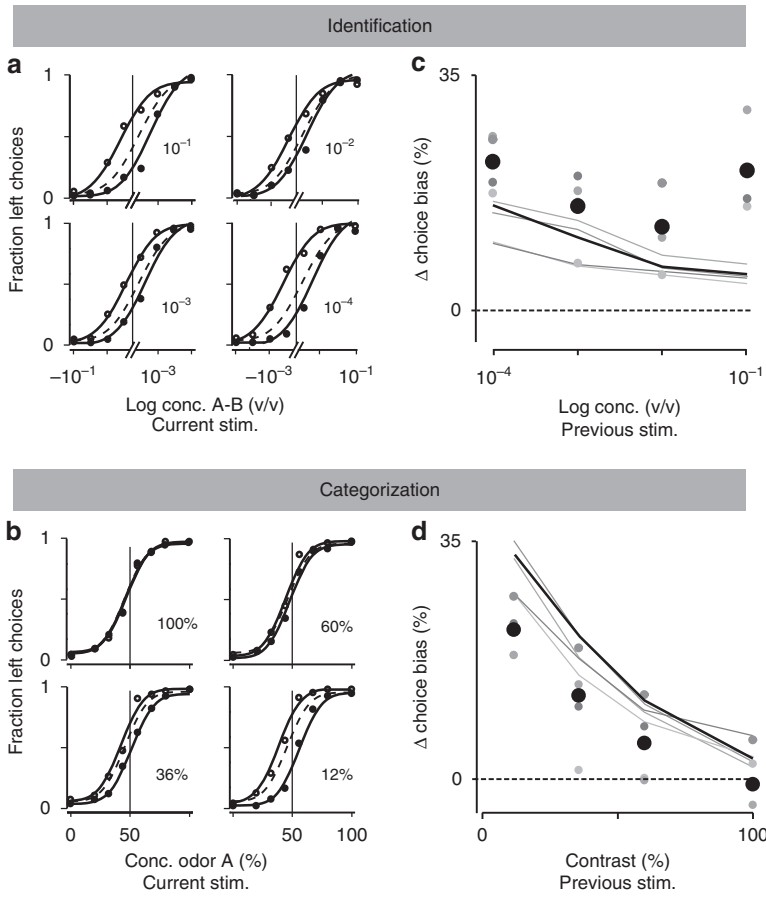

**Fig. 6 Bayes-DDM predicts trial-by-trial change of choice bias after a rewarded trial and given stimulus. a, b** Psychometric functions for mean (dashed line) and conditional curves following a reward, side choice and a given stimuli. For the identification task all four stimuli from $10^{-1}$ to $10^{-4}$ are depicted (**a**). For the categorization task from 12 to 100% mixture contrast (**b**). Filled-in circles and equivalent lines represent trials following a right-reward (B > A) and open circles, trials following a left-reward (A > B). Lines represent fits of a cumulative Gaussian to the data (Methods). **c, d** Change in choice bias plotted as a function of the previous stimuli, for plots in **a** and **b**. All four different odor concentrations for identification task (**c**); and all mixture sets for categorization (**d**). Points correspond to behavioral data (left side), and solid lines to the predicted change from the model fitted to Fig. 5 (right side). Black points and line correspond to the obtained measurements considering all data together and predicted size of effect when Bayes-DDM fits chrono- and psychometric curves for both tasks. Depicted is also each one of the four rats as points with a different shades of gray, with their analogous model predictions (gray lines).

that adjusts only the bias on a trial by trial basis, but with randomly fluctuating sensory weights (Supplementary Fig. 17). This model retained the sequential effects seen in the identification task, but failed to produce those of the categorization task (Supplementary Fig. 17g). Taken together, these modeling results suggest that the learning-induced bias fluctuations support the sequential effects in identification, whereas the learning-induced weight fluctuations support the sequential effects in categorization.

As a final model, we explored a simpler, heuristic implementation of the Bayes-DDM rule using a delta rule that is modulated by decision confidence. For this purpose, we used a standard DDM with learning rules of the form:

$$\Delta \boldsymbol{w} = \alpha \left( \lambda - \frac{\theta_{t=T}}{\theta_{t=0}} \right) \boldsymbol{s} \qquad (5)$$

$$\Delta b = \alpha_{\mathrm{b}} \left( \lambda - \frac{b}{\theta_{t=0}} \right) \qquad (6)$$

where $\theta_{t=0}$ is the value of bound at the beginning of the trial, whereas $\theta_{t=T}$ is the value of the bound at the time of the decision, $\lambda$ is the correct choice (1 or $-1$), and $\alpha$ and $\alpha_{\mathrm{b}}$ are the weight and bias learning rates. The modulation of learning by confidence is

due to the term $\frac{\theta_{t=T}}{\theta_{t=0}}$. The collapsing bound causes this ratio to decrease with elapsed time. Critically, elapsed time is inversely proportional to confidence in DDMs when the difficulty of the task is unknown and varies from trial to trial[35,48]. Therefore, for incorrect trials, the error term in this learning rule is decreasing as confidence decreases over time, which is to say the model learns less when it is less confident. Interestingly, for correct trials, the relationship is inverted, as the model learns more strongly when less confident, which makes intuitive sense: in confident correct trials there is no more information to be gained. Ultimately, this rule is only an approximation to the optimal rule. Nonetheless, Bayesian model comparison revealed that this learning rule accounts for our experimental data nearly as well as the full optimal learning rule, thus indicating that the rat's behavior is consistent with a confidence weighted learning rule (Fig. 7, Supplementary Fig. 18, RL-DDM).

The results of Bayesian model comparisons are often sensitive to the way extra parameters are penalized. We found this not to be the case in our data, as the ranking of the models remained the same whether we use AIC, AICc or BIC. Moreover, our conclusions held whether we fitted individual animals separately, or as if obtained from a single 'meta-rat' (Figs. 6, 7 and Supplementary Fig. 4).

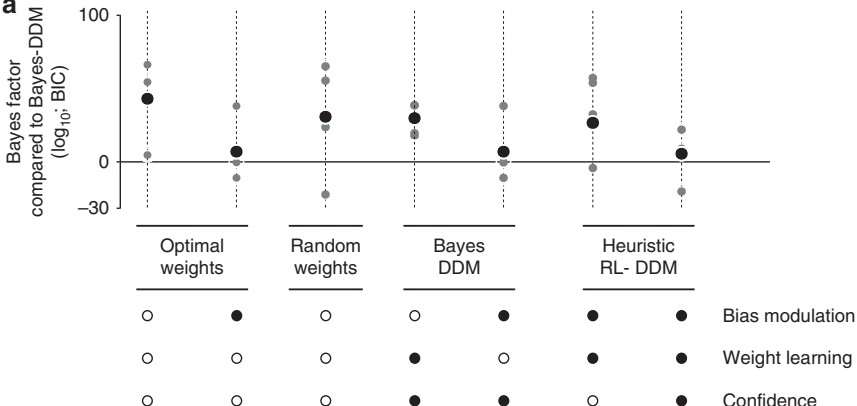

**Fig. 7 Bayesian model comparison. a** Model comparison between different family of models against optimal model (Bayes-DDM) using Bayes factor. Although the models were only fitted on the mean psycho- and chronometric function, the quality of the fitted data were evaluated considering an objective function additionally takes each model's prediction of the conditional psychometric functions regarding previous reward into account (see Methods for details). Here we depict: two optimal DDM that do not learn weights, one with trial-by-trial bias modulation and one without; a random weights model in which stimulus-to-bound weights fluctuate from trial to trial stochastically; two simplified versions of Bayes-DDM: one without bias modulation and another without weight learning; and two heuristic models in which we implement a network that mimics a DDM that learns the stimulus-to-choice map through Reinforcement learning (RL-DDM), being one of them without confidence weighted learning. The gray dots show the Bayes factor for individual rats, and the black dots the Bayes factors for the fits using the accumulated data across all rats (adjusted for the increased number of trials). For more details about these models and other versions see Methods and Supplementary material.

**Fluctuations in category boundary degrade odor categorization performance more than identification.** Finally, we sought to gain insight into how category boundary learning works in conjunction with stimulus integration to explain the difference between identification and categorization performance. To do so, we calculated "inferred" drift rates ($\mu$) for that trial by taking the actual accumulated evidence (before weight multiplication) and divided by integration time. This allowed us to visualize the combined effects of stochastic noise and boundary (weight) fluctuations (Methods) (Fig. 8). Here, for all panels we plot the evidence for the two options ($\mu_1$, $\mu_2$) against one another, so that the ideal category boundary is the diagonal (black line). In Fig. 8a, b, we show all the stimuli for the standard DDM, whereas Fig. 8c–f focus on only one of the hardest stimuli. Figure 8c, d shows the standard, non-learning DDM fit to the identification task and tested on both. Where accuracy should be similar for the two tasks, it can be seen that this model generates too few errors for the categorization task (compare the low fraction of red dots (error) to blue dots (correct) in Fig. 8d vs. c). In Fig. 8e, f, we re-ran the same trials, using "frozen noise", but simulating a fluctuating bound comparable to the Bayes-DDM with optimal learning (Methods). Here the meaning of the dot colors is different: trials that did not change classification are gray, trials that became incorrect are red, and trials that became correct are blue. It can be seen that weight fluctuations changed the classification of very few trials in the identification task (Fig. 8e) but changed a substantial fraction in the categorization task (Fig. 8f), the majority of which became errors (red).

The difference in effects on the two tasks can be understood by considering that stimulus weights have a multiplicative effect on evidence strength. Thus, stimulus weight fluctuations correspond to rotations around the origin and are larger for larger stimulus values. Therefore, high concentration mixtures, which are far from the origin, are much more susceptible to these fluctuations than low concentration stimuli (Fig. 8e). In contrast, variability in the bias, $b$, affect the intercept of the bound, giving rise to additive effects that are similar no matter the magnitude of the evidence and therefore affect the two tasks in a similar way (Supplementary Fig. 19).

## Discussion

Our results demonstrate that rats show different speed-accuracy tradeoffs (SAT) depending on the task at hand. When challenged to identify odors at low concentrations, rats show a significant increase of reaction time (RT) that is accompanied by performance degradation (Fig. 2c, d). In contrast, when the challenge is to categorize mixtures of two odors in different proportions, rats show only a small increase in RT (Fig. 2e, f). We used a standard drift-diffusion model (DDM) to show that this difference cannot be explained by stimulus noise (Fig. 3) even with the addition of reward-dependent choice biases (Supplementary Fig. 15). We therefore introduced a Bayesian learning process, the kind theorized to drive stimulus-response learning optimally in dynamic environments[32]. With the combination of these three factors—stimulus noise, reward bias and categorical boundary learning—the resulting "Bayes-DDM" not only fit the average performance data (Fig. 6b–e), but also predicted the choice biases on the recent history of stimuli, choices and rewards (Fig. 6f, g). Furthermore, Bayes-DDM was able to fit the performance over an interpolated stimulus space combining both tasks (Fig. 6d), ruling out differences in strategies between the two tasks and arguing that rats used the same decision-making system while identifying and categorizing odors.

We found that odor categorization performance is more susceptible to category boundary fluctuations than identification (Fig. 8) which in turn implies that the categorization task benefits less from longer temporal integration. Indeed, additional Bayes- and RL-DDM simulations showed that performance remains almost unaltered in mixture categorization with an increase of integration threshold, contrasting with what would be predicted for odor identification (Supplementary Fig. 20). This agrees with the observation that one sniff (the minimal unit of olfactory sampling time for animals such as a rat) is enough for maximum performance in mixture categorization[15]. Weight fluctuations, which impair performance in a trial-by-trial basis, cannot be filtered out within the integration process. On the other hand, the identification task is mostly affected by stimulus noise, which is reflected within the diffusion process, and thus benefits much more from integration. We thus conclude that the observation of

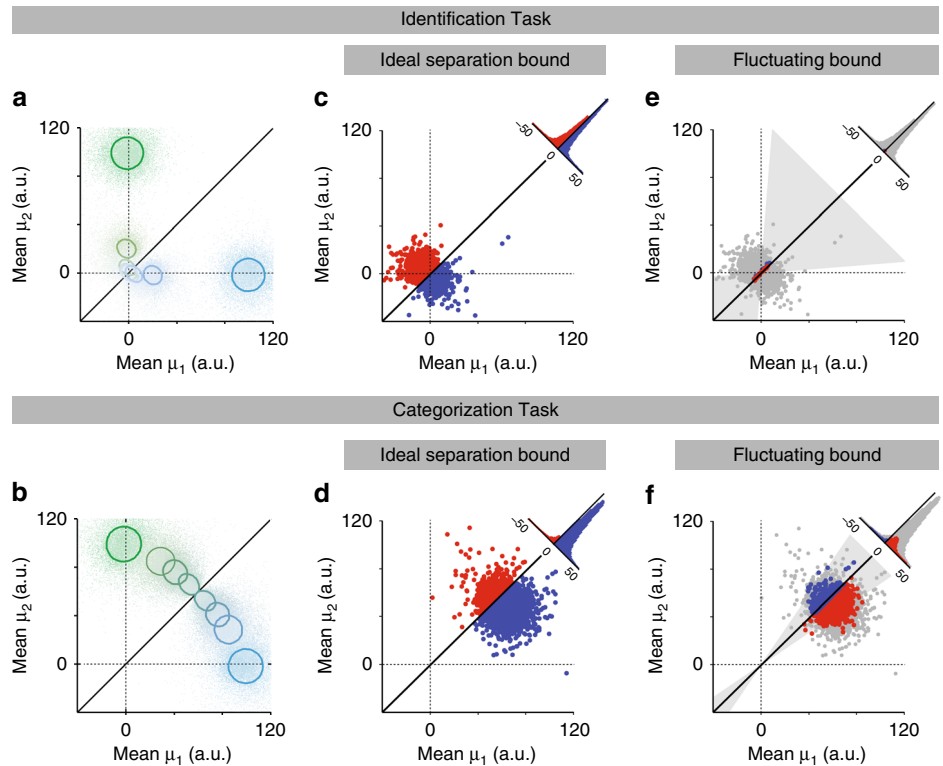

**Fig. 8 Weight fluctuations amplify errors in categorization task. a, b** Stimulus space for categorization task. Each point represents a combination of inferred drift rates for a given trial in the pure DDM (with no learning) that was fit to identification task (see Methods for more details). Solid oval lines represent the Mahalanobis distance of 1 in relation to the population average for each of the eight stimuli. Solid black line depicts the ideal classifying process: above it implies a right-side decision, below it a left. Color code for each point and line follows the same logic as Fig. 2a, b. The larger overlap of each set in the identification task (**a**) explains the performance degradation, as most points are located around the origin. For the categorization task (**b**), the lack of overlap between stimuli clarifies the higher performance seen in Fig. 4. **c, d** Mean drift rates for the most difficult left-decision choice for Bayes-DDM model (A > B, with correct choices always being left choices). In the case of the identification task this is the $10^{-4}/0$ stimuli (**c**); for the categorization task the stimulus is the 56%/44% mixture (**d**). Blue signal the correct classified choices and red the incorrect. Considering the ideal separation bound, we show the projected histograms for the difference $\langle\mu_1\rangle - \langle\mu_2\rangle$. **e, f** Same as **c, d** but now with fluctuating weights depicted as the slope of the category bound. Gray area indicates the weight fluctuations that are equivalent to 1 standard deviation for both tasks. Blue indicates trials that were originally incorrect in **c, d** but became correct, and red indicates trials that became incorrect but originally correct. Light gray dots indicate answers that remained unchanged. Histograms quantify the four populations of dots.

different SATs is due to different computational requirements in the two tasks.

Previous studies have explored dynamic stimulus learning using signal detection theory[52–58], but none has attempted to accounting for the impact of learning on evidence accumulation and RTs. Frank and colleagues combined reinforcement learning with DDMs in the RL-DDM model[50–52]. However, in this model the integration and learning processes are completely separable and did not interact. Our results show that learning can interact with evidence accumulation and that this can be detrimental for psychophysical performance. We suggest that this takes place because animals adopt a strategy that is optimized under the assumption of a dynamic environment, whereas the actual environment is static.

The continual, performance-hindering, learning we observed is striking considering that our task itself doesn't change over months of testing. This was not due to incomplete learning, as performance was stable over the analyzed data (Supplementary Figs. 21, 22). The psychophysics-like experimental paradigm is indeed highly artificial in the sense that outcomes and states are crystallized. It is unlikely that this would be the case in a more naturalistic environment, where, due to environmental dynamics, odors could signal different outcomes, rewards and states over time. A normal, ever-changing environment would imply adaptability and never-ending learning as the optimal strategy. This strategy becomes suboptimal in a static environment, but this may be a small price to pay compare to the cost of stopping learning erroneously when the world is actually dynamic. These results are consistent with a recent proposal that suboptimal inference, as opposed to internal noise, is a major source of behavioral variability[33]. In this case, the apparent suboptimal inference is the result of assuming that the world is dynamic when, in fact, it is static.

## Methods

**Animal subjects.** Four Long Evans rats (200–250 g at the start of training) were trained and tested in accordance with European Union Directive 2010/63/EU. All procedures were reviewed and approved by the animal welfare committee of the Champalimaud Centre for the Unknown and approved by the Portuguese Veterinary General Board (Direcção Geral de Veterinária, approval 0421/000/000/2019). Rats were pair-housed and maintained on a normal 12 h light/dark cycle and tested during the daylight period. Rats were allowed free access to food but were water-restricted. Water was available during the behavioral session and for 20 min after the session at a random time as well as on non-training days. Water availability was adjusted to ensure animals maintained no <85% of ad libitum weight at any time.

**Testing apparatus and odor stimuli.** The behavioral apparatus for the task was designed by Z.F.M. in collaboration with M. Recchia (Island Motion Corporation, Tappan, NY). The behavioral control system (BControl) was developed by Z.F.M., C. Brody (Princeton University) in collaboration with A. Zador (Cold Spring

Harbor Laboratory). The behavioral setup consisted of a box (27 × 36 cm) with a panel containing three conical ports (2.5 cm diameter, 1 cm depth). Each port was equipped with an infrared photodiode/phototransistor pair that registered a digital signal when the rat's snout was introduced into the port ("nose poke"), allowing us to determine the position of the animal during the task with high temporal precision. Odors were delivered from the center port and water from the left and right ports. Odor delivery was controlled by a custom made olfactometer designed by Z. F.M. in collaboration with M. Recchia (Island Motion Corporation, Tappan, NY). During training and testing the rats alternated between two different boxes.

The test odors were S-(+) and R-(−) stereoisomers of 2-octanol, chosen because they have identical vapor pressures and similar intensities. In the odor identification task, difficulty was manipulated by using different concentrations of pure odors, ranging from $10^{-4}$ to $10^{-1}$ (v/v). The different concentrations were produced by serial liquid dilution using an odorless carrier, propylene glycol (1,2-propanediol). In the odor mixture categorization task, we used binary mixtures of these two odorants at different ratios, with the sum held constant: 0/100, 20/80, 32/68, 44/56 and their complements (100/0, etc.). Difficulty was determined by the distance of the mixtures to the category boundary (50/50), denoted as "mixture contrast" (e.g., 80/20 and 20/80 stimuli correspond to 60% mixture contrast). Choices were rewarded at the left choice port for odorant A (identification task) or for mixtures A/B > 50/50 (categorization task) and at the right choice port for odorant B (identification task) or for mixtures A/B < 50/50 (categorization task). In both tasks, the set of eight stimuli were randomly interleaved within the session. During testing, the probability of each stimulus being selected was the same.

For the experiment in Figs. 2, 3, 5 and 6, only mixtures with a total odor concentration of $10^{-1}$ were used. For the experiment in Fig. 4, we used the same mixture contrasts with total concentrations ranging from $10^{-1}$ to $10^{-4}$ prepared using the diluted odorants used for the identification task. In each session, four different mixture pairs were pseudo-randomly selected from the total set of 32 stimuli (8 contrasts at 4 different total concentrations). Thus, for this task, a full data set comprised 4 individual sessions.

For all the different experiments, four of the eight stimuli presented in each session were rewarded on the left (odorant A, for identification; A/B > 50/50, for categorization) and the other four were rewarded on the right (odorant B, for identification; A/B < 50/50, for categorization). Each stimulus was presented with equal probability and corresponded to a different filter in the manifold.

For the experiments in Supplementary Fig. 3, we used two different sets of mixture ratios: 0/100, 17/83, 33.5/66.5, 50/50 in one experiment and 0/100, 39/61, 47.5/52.5, 49.5/50.5 in the second experiment. In the experiment using 50/50 mixture ratios, we used two filters both with the mixture 50/50, one corresponding to the left-rewarded stimulus and the other one to the right-rewarded stimulus. Thus, for the 50/50 mixtures, rats were rewarded randomly, with equal probability for both sides.

**Reaction time paradigm**. The timing of task events is illustrated in Fig. 1. Rats initiated a trial by entering the central odor-sampling port, which triggered the delivery of an odor with delay ($d_{odor}$) drawn from a uniform distribution with a range of [0.3, 0.6] s. The odor was available for up to 1 s after odor onset. Rats could exit from the odor port at any time after odor valve opening and make a movement to either of the two reward ports. Trials in which the rat left the odor sampling port before odor valve opening (~4% of trials) or before a minimum odor sampling time of 100 ms had elapsed (~1% of trials) were considered invalid. Odor delivery was terminated as soon as the rat exited the odor port. Reaction time (the odor sampling duration) was calculated as the difference between odor valve actuation until odor port exit (Fig. 1) minus the delay from valve opening to odor reaching the nose. This delay was measured with a photo ionization detector (mini-PID, Aurora Scientific, Inc) and had a value of 53 ms.

Reward was available for correct choices for up to 4 s after the rat left the odor sampling port. Trials in which the rat failed to respond to one of the two choice ports within the reward availability period (~1% of trials) were also considered invalid. For correct trials, water was delivered from gravity-fed reservoirs regulated by solenoid valves after the rat entered the choice port, with a delay ($d_{water}$) drawn from a uniform distribution with a range of [0.1, 0.3] s. Reward was available for correct choices for up to 4 s after the rat left the odor sampling port. Trials in which the rat failed to respond to one of the two choice ports within the reward availability period (0.5% of trials) were also considered invalid. Reward amount ($w_{rew}$), determined by valve opening duration, was set to 0.024 ml and calibrated regularly. A new trial was initiated when the rat entered odor port, as long as a minimum interval ($d_{inter-trial}$), of 4 s from water delivery, had elapsed. Error choices resulted in water omission and a "time-out" penalty of 4 s added to $d_{inter-trial}$. Behavioral accuracy was defined as the number of correct choices over the total number of correct and incorrect choices. Invalid trials (in total 5.8 ± 0.8% of trials, mean ± SEM, $n = 4$ rats) were not included in the calculation of performance accuracy or reaction times (odor sampling duration or movement time).

**Training and testing**. Rats were trained and tested on three different tasks: (1) a two-alternative choice odor identification task; (2) a two-alternative choice odor mixture categorization task[15]; and (3) a two-alternative choice "odor mixture identification" task. The same rats performed all three tasks and all other task variables were held constant.

The training sequence consisted of: (I) handling (2 sessions); (II) water port training (1 session); (III) odor port training, in which a nose poke at the odor sampling port was required before water was available at the choice port. The required center poke duration was increased from 0 to 300 ms (4–8 sessions); (IV) introduction of test odors at a concentration of $10^{-1}$, rewarded at left and right choice ports according to the identity of the odor presented (1–5 sessions); (V) introduction of increasingly lower concentrations (more difficult stimuli) (5–10 sessions); (VI) training on odor identification task (10–20 sessions); (VII) testing on odor identification task (14–16 sessions); (VIII) training on mixture categorization task (10–20 sessions); (IX) testing on mixture categorization task (14–15 sessions); (X) testing on mixture identification task (12–27 sessions) (Supplementary Fig. 1).

During training, in phases V and VI, we used adaptive algorithms to adjust the difficulty and to minimize bias of the animals. We computed an online estimate of bias:

$$b_t = (1 - \tau)C_t + \tau b_{t-1} \quad (7)$$

where $b_t$ is the estimated bias in the current trial, $b_{t-1}$ is the estimated bias in the previous trial, $C_t$ is the choice of the current trial (0 if right, 1 if left) and $\tau$ is the decay rate ($\tau = 0.05$ in our experiments). The probability of being presented with a right-side rewarded odor $p$ was adjusted to counteract the measured bias using:

$$p_R = 1 - \frac{1}{1 + e^{\frac{(b_t - b_0)}{\gamma}}} \quad (8)$$

where $b_0$ is the target bias (set to 0.5), and $\gamma$ (set to 0.25) describes the degree of non-linearity.

Analogously, the probability of a given stimulus difficulty was dependent on the performance of the animal, i.e., the relative probability of difficult stimuli was set to increase with performance. Performance was calculated in an analogous way as (1) at the current trial but $c_t$ became $r_t$—the outcome of the current trial (0 if error, 1 if correct). A difficulty parameter, $\delta$, was adjusted as a function of the performance,

$$\delta_{t+1} = -1 + \frac{2}{1 + e^{\frac{(p_t - p_0)}{\gamma}}} \quad (9)$$

where $p_0$ is the target performance (set to 0.95). The probability of each stimulus difficulty, $\varphi$, was drawn from a geometric cumulative distribution function (GEOCDF, Matlab)

$$\varphi_{t+1} = \frac{1 - \mathrm{GEOCDF}(i, |\delta_{t+1}|)}{\sum_{j=1}^{N} 1 - \mathrm{GEOCDF}(j, |\delta_{t+1}|)} \quad (10)$$

where $N$ is the number of stimulus difficulties in the session, and takes a value from 2 to 4 (when $N = 1$, i.e. only one stimulus difficulty, this algorithm is not needed); $i$ corresponds to the stimulus difficulty and is an integer from 1 to 4 (when $\delta > 0$, the value 1 corresponds to the easiest stimuli and 4 to the most difficult one, and vice-versa when $\delta < 0$). In this way, when $|\delta|$ is close to 0, corresponding to an average performance close to 0.95, the distribution of stimuli was close to uniform (i.e. all difficulties are equally likely to be presented). When performance is greater, then the relative probability of difficult trials increased; conversely, when the performance is lower, the relative probability of difficult trials decreased. Training phases VI and VIII were interrupted for both tasks when number of stimulus difficulties $N = 4$ and difficulty parameter $\delta$ stabilized on a session-by-session basis.

Each rat performed one session of 90–120 min per day (250–400 trials), 5 days per week for a period of ~120 weeks. During testing, the adapting algorithms were turned off and each task was tested independently. The data set was collected only after performance was stable (Supplementary Fig. 21) during periods in which the animals showed stable accuracy and left/right bias on both tasks (Supplementary Figs. 21, 22). Throughout the test period, there was variability in accuracy and bias across sessions, but there was no correlation between these performance metrics and session number (accuracy: Spearman's rank correlation $\rho = -0.066$, $P = 0.61$ for identification, $\rho = 0.16$, $P = 0.24$ for categorization; bias: $\rho = 0.104$, $P = 0.27$ for both tasks, identification: $\rho = 0.093$, $P = 0.48$, categorization: $\rho = 0.123$, $P = 0.37$).

**Session bias and choice bias**. The psychometric curves are obtained by fitting the function, $\psi(x)$:

$$\psi(x) = l^L + (1 - l^L - l^R)\Phi(x|\mu, \sigma) \quad (11)$$

where $x$ is the stimulus, $\Phi(x|\mu,\sigma)$ is the cumulative function of a Gaussian with mean and variance, $\mu$ and $\sigma$, and $l^L$ and $l^R$ are the right and left lapse rates. The parameters are fitted by minimizing the square distance between $\psi(x)$ and the empirical fraction of right-ward choices for each stimulus value $x$ through fmin-search (Matlab)[28,37]:

To quantify how a reward and its interaction with stimulus difficulty impacts choice bias, we also fitted psychometric curves for the current trial $T$, conditioned on the response and difficulty of the previous trials, $T-1$:

$$\psi(x_T | C_{T-1} = R, d_{T-1}) = l_T^L + (1 - l_T^L - l_T^R)\Phi(x_T|\mu_T, \sigma_T; C_{T-1} = R, d_{T-1})$$

where $C_{(T-1)} = R$ means that the animal made a correct, right-ward choice on the previous trial. The difficulty variable $d_{(T-1)}$ indicates that the stimulus in the

previous trial took the value $x_{(T-1)}$ or $x^{\dagger}_{T-1}$ corresponding to difficulty level $d_{(T-1)}$. For instance, in the identification task, the conditions in which $[A]=10^{-1}$ or $[B]=10^{-1}$ corresponds to the same difficult level. Difficulty corresponds to the x-axis on Fig. 6c–d and Supplementary Figs. 7 and 8).

We then quantified the change in choice bias as a function of the difficulty on the previous trial as follow:

$$\Delta C^c_b(d_{T-1}) = \frac{1}{2}\left(\psi(x_T = I|C_{T-1} = R, d_{T-1}) - \psi(x_T = I|C_{T-1} = L, d_{T-1})\right) \quad (12)$$

where $I$ is the indifference point of the unconditioned psychometric curve in Eq. 11, that is, the stimulus value for which the rat chooses left or right with equal probability, $\psi(x = I) = 0.5$ (Fig. 6a–b, solid black line).

For trials following an error, we first define the following psychometric curve:

$$\psi(x_{T-2}|F_{T-1} = R, d_{T-1}) = l^L_{T-2} + \left(1 - l^L_{T-2} - l^R_{T-2}\right)$$
$$\Phi\left(x_{T-2}|\mu_{T-2}, \sigma_{T-2}, F_{T-1} = R, d_{T-1}\right)$$

where $F_{(T-1)} = R$ refers to an incorrect, right-ward response in the previous trial. This expression corresponds to the psychometric curve for two trials back but conditioned on the response and difficulty one trial back. We then define the change in choice bias as:

$$\Delta C^f_b(d_{T-1}) = \frac{1}{2}\left(\Delta C^f_b(F_{T-1} = R, d_{T-1}) - \Delta C^f_b(F_{T-1} = L, d_{T-1})\right) \quad (13)$$

where

$$\Delta C^f_b(F_{T-1} = R, d_{T-1}) = \psi(x_T = I|F_{T-1} = R, d_{T-1}) - \psi(x_{T-2} = I|F_{T-1} = R, d_{T-1})$$

$$\Delta C^f_b(F_{T-1} = L, d_{T-1}) = \psi(x_T = I|F_{T-1} = L, d_{T-1}) - \psi(x_{T-2} = I|F_{T-1} = L, d_{T-1})$$

where I is the same indifference point as for choice biases after correct trials. We here conditioned on two trials back to avoid biases introduced by long bouts of incorrect trials. For correct trials, our results are qualitatively similar, irrespective of whether we used Eq. 12 or Eq. 13. For error trials, the use of Eq. 13 over Eq. 12 had a major impact on the bias estimates and revealed a win stay loose shift strategy for our rats.

## Model

*Drift-diffusion model for decision-making.* For a given stimulus with concentrations $c_A$ and $c_B$, we define the accumulated evidence at time $t$, $e(t)$. The diffusion process has the following properties: at time $t = 0$, the accumulated combined evidence is zero, $e(0) = 0$; and the momentary evidence $m_i$ is a random variable that is independent at each time step. We discretize time in steps of 0.1 ms and run numerical simulations of multiple runs/trials. For each new time step $t = n\Delta t$, we generate a new momentary evidence draw:

$$m_i(t) = m_i(n\Delta t) = N\left(kc^{\beta}_i, \sigma\right) \quad (14)$$

that is, through a normally distributed random generator with a mean of $kc^{\beta}_i$, in which we define $k$ as the sensitive parameter, and $\beta$ as the exponent parameter. $\sigma$ defines the amount of noise in the generation of momentary evidences. We set $\sigma$ to 1, making $kc^{\beta}_i$ equivalent to the signal to noise ratio for a particular stimuli and respective combination of concentrations ($c_A$, $c_B$). Integrated evidences ($s_1$, $s_2$) are simply the integration of the momentary evidences over time

$$s_i(t) = \int^t_{\tau=0} m_i(\tau)d\tau \quad (15)$$

We translate this in our discretized version as a cumulative sum at all time steps, effectively being:

$$s_i(n\Delta t) = \sum^n_{j=0} m_i(j\Delta t) \quad (16)$$

We then define the decision variable accumulated evidence as:

$$e(t) = w_1 s_1(t) + w_2 s_2(t) + b \quad (17)$$

or in its discretized version:

$$e(n\Delta t) = w_1 s_1(n\Delta t) + w_2 s_2(n\Delta t) + b \quad (18)$$

where $w_1$ and $w_2$ are model-dependent combination weights on the accumulated evidence, and $b$ is an a priori decision bias ($w_1 = 1/\sqrt{2}$; $w_2 = -1/\sqrt{2}$; $b = 0$ for optimal decisions; $\sqrt{2}$ scaling ensures $||w|| = 1$). Together, these parameters define slope and offset of the category boundary, which determines the mapping between accumulated evidence and associated choices. We also define the (accumulation) decision bound $\theta(t)$ and make it in most models collapsing over time through either a linear or an exponential decay. Thus, at time step $n\Delta t$ the bound is either

$$\theta(t) = \theta(n\Delta t) = \theta_{t=0} + \theta_{slo}n\Delta t \quad (19)$$

where we define $\theta_{t=0}$ as the bound height at the starting point of integration $t = 0$ and $\theta_{slo} \leq 0$ as its slope, or

$$\theta(t) = \theta(n\Delta t) = \theta_{t=0}e^{-n\Delta t/\tau} \quad (20)$$

where $\tau \geq 0$ is the bound height's mean lifetime. The collapse parameters $\theta_{slo}$ and $\tau$ define the level of urgency in a decision, the smaller it becomes, the more urgent a given decision will become, given rise to more errors[35,42]. For models with non-collapsing boundaries, we used $\theta(t) = \theta_{t=0}$, independent of time. For models with collapsing boundaries, they collapsed linearly, except for RL-DDM, where they collapse exponentially.

Decisions are triggered once the accumulated evidence, $e(t)$, crosses one of the two decision boundaries $\{\theta(t), -\theta(t)\}$. To simulate these decisions, we first simulated a one-dimensional diffusion model that directly uses $e(t)$ as the diffusing "particle", and from this reconstructed the higher-dimensional accumulated momentary evidences $s(t) = (s_1(t), s_2(t))^T$. For the one-dimensional simulation, we used a momentary Gaussian evidence with drift $w_1 kc^{\beta}_1 + w_2 kc^{\beta}_2$ and diffusion variance $w^2_1 + w^2_2$ (both per unit time step), corresponding to the moments of $e(t) - b$. We reintroduce the bias $b$ by shifting the boundaries to $\{\theta(t) - b, -\theta(t) - b\}$. For non-collapsing boundaries, we simulated accumulation boundary crossings using a recently developed, fast, and unbiased method[53]. For collapsing boundaries, we simulated these boundary crossing by Euler integration in $\Delta t = 0.001s$ time steps, and set the final $e(t)$ to lie on the crossed boundary to avoid overshooting that might arise due to time discretization. In both cases, we defined the decision time $t_d$ as the time when crossing occurred, and the choice in trial $k$ by

$$C_k = \text{choice} = \{\text{left}, e(t_d) > 0 \text{ right}, e(t_d) < 0\} \quad (21)$$

To recover the higher-dimensional accumulated momentary evidences at decision time, $s(t_d)$, we sampled those from the two-dimensional Gaussian $s(t_d)|e(t_d), t_d \sim N\left((c^{\beta}_1, c^{\beta}_2)^T k t, I t\right)$ (i.e., unbounded diffusion), subject to the linear decision boundary constraint $w^T s(t_d)\bar{s} + b = e(t_d)$, using the method described in ref. [54].

In order to capture <100% accuracy in easy trials and systematic and consistent choice biases, we introduced an additional "lapse" component with lapse rate $l_r$ and bias b to the model. The lapse rate $l_r$ determined the probability with which the choice is not determined by the diffusion model, but is instead drawn from a Bernoulli distribution that chooses "right" with probability $l_r$ and "left" with probability $1 - l_r$. Model fits revealed small lapse rates close to 0.05 (Supplementary Tables 1 and 2). These lapse rates are typically needed for this type of models and have been hypothesized in the past to be due to effects of attention and/or exploration[55].

Lastly, the reaction time for a particular trial was simulated by adding a normally distributed non-decision time variable with mean $t_{ND}$ and standard deviation $0.1t_d$ to the decision time arising from the diffusion model simulations[2],

$$t_r = t_d + t_{ND} + \eta_{ND}, \quad (22)$$

where $\eta_{ND}|t_d \sim N(0, (0.1 t_d)^2)$ models the stochasticity of the non-decision time. Without weight and bias learning (that is, when fixing $w_1 = 1/\sqrt{2}$; $w_2 = -1/\sqrt{2}$; $b = 0$), the base model with a non-collapsing has the following six parameters: sensitivity ($k$), exponent ($\beta$), non-decision time mean ($t_{ND}$), initial bound height ($\theta_{t=0}$), lapse rate ($l_r$), and bias ($b$). A collapsing bound introduces one additional parameter, which is the boundary slope ($\theta_{slo}$) for linearly collapsing boundaries, or the boundary mean lifetime ($\tau$) for exponentially collapsing boundaries.

**Drift-diffusion model with Bayesian reward bias and stimulus learning—Bayes-DDM.** The following provides an overview of the Bayesian model that learns stimulus combination weights, reward biases, or both. A complete description of the model and its derivation can be found in ref. [32]. We first focus on weight learning, and then describe how to apply the same principles to bias learning. The model assumes that there are true, latent combination weights $w^*$ that the decision-maker cannot directly observe, but aims to infer based on feedback on the correctness of his/her choices. To ensure continual learning, these latent weights are assumed to slowly change across consecutive trials $k$ and $k + 1$ according to a first-order autoregressive process,

$$w^{*(k+1)}|w^{*(k)} \sim N\left(\gamma_w w^{*(k)}, \sigma^2_w I\right), \quad (23)$$

with weight "leak" $0 \leq \gamma_w < 1$, ensuring that weights remain bounded, and weight diffusion variance $\sigma^2_w$, ensuring a continual, stochastic weight change. This process has zero steady-state mean and a steady-state variance of $\sigma^2_w/(1 - \gamma^2_w)$ for each of the true weight components, which we used as the decision-maker's prior $p(w)$ over the inferred weight vector $w$.

For each sequence of trials that we simulated, the decision-maker starts with this prior in the first trial and updates its belief about the weight vector in each subsequent trial in two steps. We describe these two steps in light of making a choice in trial $k$, receiving feedback about this choice, updating one's belief, and then moving on to the next trial $k + 1$. Before the first step in trial $k$, the decision-maker holds the "prior" belief $p(w^{(k)}|\text{all past information}) = p(w^{(k)})$ that is implicitly conditional on all feedback received in previous trials $1, \cdots, k - 1$. The decision-maker then observes some sensory evidence, accumulates this evidence, commits to choice $C_k$ with decision time $t_k$ and accumulated momentary evidences $s(t_d)$. After this, the correct choice $C^*_k \in \{-1, 1\}$ (-1 for "left", 1 for "right") is

revealed, which, in our 2-AFC setup is the same as telling the decision-maker if choice $C_k$ was correct or incorrect. The Bayes-optimal way to update one's belief about the true weights upon receiving this feedback is given by Bayes' rule,

$$p\left(\boldsymbol{w}^{(k)}|C_k^*, \boldsymbol{s}(t_d), t_d\right) \propto p\left(C_k^*|\boldsymbol{w}^{(k)}, \boldsymbol{s}(t_d), t_d\right) p\left(\boldsymbol{w}^{(k)}\right). \quad (24)$$

Unfortunately, the functional form of the likelihood $p\left(C_k^*|\boldsymbol{w}^{(k)}, \boldsymbol{s}(t_d), t_d\right)$ does not permit efficient sequential updating of this belief, but we have shown elsewhere[32] that we can approximate the above without considerable performance loss by assuming that the posterior (and, by induction, also the prior) is Gaussian. Using prior parameters $p\left(\boldsymbol{w}^{(k)}\right) = N\left(\boldsymbol{w}^{(k)}\big|\boldsymbol{\mu}_w^{(k)}, \Sigma_w^{(k)}\right)$ and posterior parameters $p\left(\boldsymbol{w}^{(k)}|C_k^*, \boldsymbol{s}(t_d), t_d\right) = N(\boldsymbol{w}^{(k)}|\boldsymbol{\mu}_w^{+(k)}, \Sigma_w^{\prime(k)})$ yields the update equations

$$\boldsymbol{\mu}_w^{+(k)} = \boldsymbol{\mu}_w^{(k)} + \alpha_w(\boldsymbol{s}(t_d), t_d) C_k^* \Sigma_w^{(k)} \boldsymbol{s}(t_d) \quad (25)$$

$$\Sigma_w^{\prime(k)} = \Sigma_w^{(k)} + \alpha_{\text{cov}}(\boldsymbol{s}(t_d), t_d)\left(\left(\Sigma_w^{(k)^{-1}} + \tilde{\boldsymbol{s}}\tilde{\boldsymbol{s}}^T\right)^{-1} - \Sigma_w^{(k)}\right), \quad (26)$$

with learning rates

$$\alpha_w(\boldsymbol{s}(t_d), t_d) = \frac{N(g|0, 1)}{\Phi(g)\sqrt{1 + \tilde{\boldsymbol{s}}^T \Sigma_w^{(k)} \tilde{\boldsymbol{s}}}}, \quad (27)$$

$$\alpha_{\text{cov}}(\boldsymbol{s}(t_d), t_d) = \alpha_w(\boldsymbol{s}(t_d), t_d)^2 + \alpha_w(\boldsymbol{s}(t_d), t_d) g, \quad (28)$$

$$g = \frac{C_k^* \boldsymbol{\mu}_w^{(k)^T} \tilde{\boldsymbol{s}}}{\sqrt{1 + \tilde{\boldsymbol{s}}^T \Sigma_w^{(k)} \tilde{\boldsymbol{s}}}}, \quad (29)$$

$$\tilde{\boldsymbol{s}} = \frac{\boldsymbol{s}(t_d)}{\sqrt{t_d + \sigma_e^{-2}}}, \quad (30)$$

where $\Phi(\cdot)$ is the cumulative function of a standard Gaussian, and where $\sigma_e^2$ is a variance that describes the distribution of decision difficulties (e.g., odor intensities) across trials, and which we assume to be known by the decision-maker. In the above, $g$ turns out to be a quantity that is closely related to the decision confidence in trial $k$. Furthermore, both learning rates, $\alpha_w$ and $\alpha_{\text{cov}}$ are strongly modulated by this confidence, as follows: they are small for high-confidence correct decisions, moderate for low-confidence decisions irrespective of correctness, and high for high-confidence incorrect choices. A detailed derivation, together with more exploration of how learning depends on confidence is provided in ref. [32].

Once the posterior parameters have been computed, the second step follows. This step takes into account that the true weights change across consecutive trials, and is Bayes-optimally captured by the following parameter updates:

$$\boldsymbol{\mu}_w^{(k+1)} = \gamma_w \boldsymbol{\mu}_w^{+\ (k)}, \quad (31)$$

$$\Sigma_w^{(k+1)} = \gamma_w^2 \Sigma_w^{\prime(k)} + \sigma_w^2 \mathbf{I}. \quad (32)$$

These parameters are then used in trial $k + 1$. Overall, the Bayesian weight learning model has two adjustable parameters (in addition to those of the base decision-making model): the assumed weight leak ($\gamma_w$) and weight diffusion variance ($\sigma_w^2$) across consecutive trials.

Let us now consider how similar principles apply to learning the bias term. For this, we again assume a true underlying bias $b^*$ that changes slowly across consecutive trials according to

$$b^{*(k+1)}|b^{*(k)} \sim N\left(\gamma_w b^{*(k)}, \sigma_b^2\right), \quad (33)$$

where the leak $\gamma_w$ is the same as for $\boldsymbol{w}^*$, but the diffusion $\sigma_b^2$ differs. As we show in ref. [32], the bias can be interpreted as a per-trial a priori bias on the correctness on either choice, which brings it into the realm of probabilistic inference. More specifically, this bias can be implemented by extending the, until now two-dimensional, accumulated momentary evidences $\boldsymbol{s}(t_d)$ in each trial, by an additional, constant element. An analogous extension of $\boldsymbol{w}$ adds the bias term to them, until now two-dimensional, weight vector. Then, we can perform the same Bayesian updating of the, now three-dimensional, weight vector parameters as described weights, to learn weights and the bias simultaneously. The only care we need to take is to ensure that, in the second step, the covariance matrix elements associated with the bias are updated with diffusion variance $\sigma_b^2$ rather than $\sigma_w^2$. Overall, a Bayesian model that learns both weights and biases has three adjustable parameters: the assumed weight and bias leak ($\gamma_w$), the weight diffusion variance ($\sigma_w^2$), and the bias diffusion variance ($\sigma_b^2$). A Bayesian model that only learns the bias has two adjustable parameters: the assumed bias leak ($\gamma_w$), and the bias diffusion variance ($\sigma_b^2$).

**Drift-diffusion model with heuristic reward bias and stimulus learning—RL-DDM.** Rather than using the Bayesian weight and bias update equations in their full complexity, we also designed a model that captures their spirit, but not their details. This model does not update a whole distribution over possible weights and biases, but instead only works with point estimates, which take values $\boldsymbol{w}^{(k)}$ and $b^{(k)}$

in trial $k$. After feedback $C_k^* \in \{-1, 1\}$ (as before, $-1$ for "left", 1 for "right"), the model updates the weight according to

$$\boldsymbol{w}^{+(k)} = \boldsymbol{w}^{(k)} + \alpha\left(C_k^* - \frac{e(t_d)}{\theta_{t=0}}\right)\boldsymbol{s}(t_d), \quad (34)$$

where $\alpha$ is the learning rate. Note that, for rapid decisions (i.e., $t_d \approx 0$), we have $|e(t_d)| \approx \theta_{t=0}$, such that the residual term in brackets is zero for correct choices, such that learning only occurs for incorrect choices. For slower choices and collapsing boundaries, we will have $|e(t_d)| < \theta_{t=0}$, such that the residual will be non-zero even for correct choices, promoting weight updates for both correct and incorrect choices. Considering that decision confidence in the Bayesian model is generally lower for slower choices, this learning rule again promotes learning rates weighted by confidence: fast, high-confidence choices result in no weight updates for correct choices, and large weight updates for incorrect choices, whereas low, low-confidence choices promote moderate updates irrespective of the correctness of the choice, just as for the Bayes-optimal updates. To ensure a constant weight magnitude, the weights are subsequently normalized by

$$\boldsymbol{w}^{(k+1)} = \frac{\boldsymbol{w}^{+(k)}}{\|\boldsymbol{w}^{+(k)}\|}, \quad (35)$$

to form the weights for trial $k + 1$.

Bias learning takes a similar flavor, using the update equation

$$b^{(k+1)} = b^{(k)} + \alpha_b\left(C_k^* - \frac{b^{(k)}}{\theta_{t=0}}\right), \quad (36)$$

where $\alpha_b$ is the bias learning rate. In contrast to weight learning, this update equation does not feature any confidence modulation, but was nonetheless sufficient to capture the qualitative features of the data. Overall, this learning model added two adjustable parameters to the base decision-making model: the weight learning rate ($\alpha$), and the bias learning rate ($\alpha_b$).

**Alternative learning heuristics.** To further investigate whether a confidence-modulated learning rate was required, we designed models that did not feature such confidence weighting. For weight learning, they used the delta rule

$$\boldsymbol{w}^{+(k)} = \boldsymbol{w}^{(k)} + \alpha\left(C_k^* - C_k\right)\frac{\theta(t_d)}{\theta_{t=0}}\boldsymbol{s}(t_d), \quad (37)$$

where $\alpha$ is the learning rate, and whose weight updated is, as before, followed by the normalization $\boldsymbol{w}^{(k+1)} = \boldsymbol{w}^{+(k)}/\|\boldsymbol{w}^{+(k)}\|$. Here, we assume the same encoding of make choice $C_k$ and correct choice $C_k^*$, that is, $C_k \in \{-1, 1\}$ ($-1$ for "left", 1 for "right"), such that the residual in brackets is only non-zero if the choice was incorrect. In that case, the learning rate is modulated by boundary height, but no learning occurs after correct choices.

The bias is learned similarly, using

$$b^{(k+1)} = b^{(k)} + \alpha\left(C_k^* - C_k\right)\frac{\theta(t_d)}{\theta_{t=0}}. \quad (38)$$

Overall, this results in one adjustable parameter in addition to the base decision-making model: the learning rate ($\alpha$).

**Drift-diffusion model with reward bias and stimulus weight fluctuations.** To test whether random weight and bias fluctuations are sufficient to capture the across-task differences, we also fit a model that featured such fluctuations without attempting to learn these weights from feedback. Specifically, we assumed that, in each trial, weights and biases where drawn from

$$\boldsymbol{w}^{(k)} \sim N\left((1, -1)^T/\sqrt{2}, \sigma_{rw}^2 \mathbf{I}\right), \ b^{(k)} \sim N\left(0, \sigma_{rb}^2\right), \quad (39)$$

which are normal distributions centered on the optimal weights and bias values, but with (co)variances $\sigma_{rw}^2 I$ and $\sigma_{rb}^2$. We adjusted these (co)variances to best match the data, leading to two adjustable parameters in addition to those of the base decision model: the weight fluctuation variance ($\sigma_{rw}^2$) and the bias fluctuation variance ($\sigma_{rb}^2$).

**Leaky, competing accumulator model—LCA.** To test whether a non-learning two-component race model with mutual inhibition is able to fit both tasks with the same set of parameters, we implemented a leaky, competing accumulator model[49]. In this model, two accumulators, $s_1$ and $s_2$ accumulate evidence according to

$$ds_i = \left(kc_i^\beta - \tau^{-1}s_i - w_{\text{inh}}s_{3-i}\right)dt + dW_i, \quad (40)$$

where $\tau$ is the leak time constant, $w_{\text{inh}}$ is the mutual inhibition weight, and $W_i$ is a Wiener process. The accumulators start at $s_1(0) = s_2(0) = 0$, are lower-bounded by $s_1(t) \geq 0$ and $s_2(t) \geq 0$, and accumulate evidence until the first of the two reaches the decision threshold $\theta(t)$, triggering the corresponding choice. The model was simulated by Euler integration in 0.1 ms time steps, and lapses and biases were implemented as for the other model. We fitted two variants, one with a time-invariant boundary, $\theta(t) = \theta_0$, with a total of eight parameters, and one with a time-variant boundary, $\theta(t) = \theta_0 + \theta_{\text{slo}}t$, with nine parameters.

**Model fitting**. We found the best-fitting parameters for each model by log-likelihood maximization[2]. Due to collapsing bounds and (for some models) sequential updates of weights and biases, we could not directly use previous approaches that rely on closed-form analytical expressions[19] for fitting diffusion models with non-collapsing boundaries. Instead, for any combination of parameters, we simulated the model responses to a sequence of 100,000 trials with stimulus sequence statistics matching those of the rodent experiments for the conditions that we were interested in fitting. Please see "Drift-diffusion model for decision-making" for details on how these simulations were performed. The simulated responses were used to compute summary statistics describing model behavior, which were subsequently used to evaluate the log-likelihood of these parameters in the context of the animals' observed behavior. We computed the log-likelihood in two ways, first by ignoring sequential choice dependencies, and second by taking such dependencies into account. All model simulations were performed as described further above. We did not explicitly simulate the stochasticity of the non-decision time, but instead included this stochasticity as an additional noise-term in the likelihood function (not explicitly shown below).

To describe how we computed the likelihood of model parameters $\phi$ without taking sequential dependencies into account, let index $m$ denote the different task conditions (i.e., a set of odor concentrations for odors A and B), and let $n_m$ be the numbers of observed trials in the rodent data that we are modeling. For each condition $m$, we approximate the response time distributions by Gaussians, using $t_m$ and $\sigma^2_{t,m}$ to denote the mean response time and variance observed in the animals' behavior (across trials). Furthermore, let $P_{c,m}$ be the observed probability of making a correct choice in that condition. The corresponding model predictions for parameters $\phi$, extracted from model simulations, are denoted $\bar{t}_m(\phi)$ and $\bar{P}_{c,m}(\phi)$. With this, we computed the likelihood of responses times by

$$L_{t,m}(\phi) = N\left(\bar{t}_m(\phi)|t_m, \frac{\sigma^2_{t,m}}{n_m}\right), \qquad (41)$$

which is the probability of drawing the predicted mean reaction time from a Gaussian centered on the animals' observed mean and with a variance that corresponds to the standard error of that mean. The likelihood of the choice probabilities was for each condition computed by

$$L_{c,m}(\phi) = \bar{P}_{c,m}(\phi)^{P_{c,m} n_m}\left(1 - \bar{P}_{c,m}(\phi)\right)^{(1-P_{c,m})n_m}, \qquad (42)$$

which is the probability of drawing the animals' observed number of correct and incorrect choices with the choice probabilities predicted by the model. The overall log-likelihood is found by summing over the per-condition log-likelihoods, resulting in

$$LL(\phi) = \sum_m \left(\log L_{t,m}(\phi) + \log L_{c,m}(\phi)\right). \qquad (43)$$

To evaluate the log-likelihood that takes into account sequential choice dependencies, we computed the reaction time likelihoods, $L_{t,m}(\theta)$, as before, but changed the choice probability likelihood computation as follows. For trials following correct choices, we computed the choice probability likelihood separately for each stimulus combination given the previous and the current trial, thus taking into account that psychometric curves depend on the stimulus condition of the previous trial (Fig. 6a, b). Due to the low number of incorrect trials for certain conditions, we did not perform this conditioning on the previous trial's condition when computing the choice probability likelihoods after incorrect choices, but instead computed the likelihood across all trials simultaneously.

For both ways of computing the log-likelihood, we found the parameters that maximize this log-likelihood by use of the Subplex algorithm as implemented in the NLopt library (Steven G. Johnson, The NLopt nonlinear-optimization package, http://ab-initio.mit.edu/nlopt). In some cases, we performed the fits without taking into account the sequential choice dependencies, and then predicted these sequential choice dependencies from the model fits (e.g., Fig. 6c, d). In other cases (e.g., for some model comparisons), we performed the model fits while taking into account sequential dependencies. The specifics of the model fits are clarified in the main text. The best model fits and respective parameters can be found in Supplementary Tables 1 and 2.

**Model comparison**. For comparison between different models with different number of parameters, we use Bayesian information criterion (BIC) for model selection[56]. For each model, we calculate the BIC[57]:

$$\text{BIC} = -2\ln(L) + q\Delta\ln(n) \qquad (44)$$

where $q$ is the number of free parameters fitted by the model and $n$ the number of trials that we fitted. Each model has a BIC associated to it. We compared different models by first converting the BIC score into a log10-based marginal likelihood, using $-0.5\text{BIC}/\ln(10)$, and then compared models by computing the log10-Bayes factor as the difference between these marginal likelihoods. These differences dictate the explanatory strength of one model in relation to the other. The model

with the larger marginal likelihood is preferred and the evidence in favor is decisive if the log10 difference exceeds 2.

To ensure that our analysis is not driven by the strong parameter number penalty that BIC applies, we performed the same analysis using the Akaike information criterion (AIC) and its corrected version (AICc), but found qualitatively no change in the results. All different model comparisons can be found in Supplementary Fig. 4.

In Supplementary Fig. 4, we compared the following models. Models denoted simply "DDM" were diffusion models with optimal weights, $w_1 = 1/\sqrt{2}$; $w_2 = -1/\sqrt{2}$. Models denoted "Bayes-DDM" learned their weights as described in the Bayes-DDM section. The "Random weights" models used weights that were stochastically and independently drawn in each trial (see Stimulus weight fluctuations section). The "Delta rule" models learned their weights by the delta rule. The "Full RL-DDM" model used the learning rules described in the RL-DDM section. Only "lapse" variants of these models included the lapse model components. Decision boundaries were constant except for the "collapsing boundary" model variants. The bias was fixed to $b = 0$, except for the "Full RL-DDM" model and "bias" variants. In these bias variants, the biases (but not necessarily the weights, depending on the model) were learned as described in the Bayes-DDM section, except for the "Delta rule" models, for which bias learning was described in the Alternative learning heuristics section. In Supplementary Fig. 4, all models are compared to the Bayes-DDM model that learns both weights and the bias, includes a lapse model, and has collapsing boundaries.

**Weights fluctuation analysis**. As the Bayes-DDM model reaches a decision, it has access to two variables, amount of evidence at the bound and the decision time $t_d$. For better understanding the dynamics immediately before the multiplication of the weights, we looked at the combination of sensory evidence ($s_1$, $s_2$) for each simulated trial. For each trial $j$, there is a noisy sensory evidence trajectory (integration layer from Fig. 6). This means that by the end of trial $j$, we can compute the mean drift rates that gave to rise to a decision:

$$\langle \mu_i^j \rangle = \frac{s_i^j}{t_d^j - t_r} \qquad (45)$$

Each group in Fig. 8a, b has been segregated taking into account the Mahalanobis distance, as each line represents the distance of $D = 1$ for a particular stimulus set.

Considering the integrated evidence of Eq. 13 and combined with the choice function of Eq. 21 we see that

$$w_1 s_1(t) + w_2 s_2(t) + b = 0 \qquad (46)$$

Should represent the separation line between the two stimuli, and thus we can rewrite Eq. 46 as:

$$s_2(t_d) = -\frac{w_1}{w_2}s_1(t_d) + \frac{b}{w_2} \qquad (47)$$

Considering the straight-line equation $y = mx + i$, we see that in our integrated evidence plots the boundary separation can be drawn with slope $m = -\frac{w_1}{w_2}$ and intercept $i = \frac{b}{w_2}$.

Stimulus weight fluctuation should then have an impact in the slope of the boundary line separating the classification between left and right stimuli, and $b$ should influence the origin intercept on that stimulus representation (Fig. 8). Considering the data points simulated for 100,000 trials, we analyzed the effect of slope fluctuation in error rates. That is, how many errors would the model create by having a particular value of $m$, for both the identification and categorization task (Fig. 8).

**Analysis**. All the behavioral and statistical analysis, as well as all fitting, were performed in Matlab®. The different models were implemented and fitted in Julia v1.0.4.

**Reporting summary**. Further information on research design is available in the Nature Research Reporting Summary linked to this article.

## Data availability
The data that support the findings of this study are available from the corresponding author upon reasonable request.

## Code availability
The code that support the findings of this study are available from the corresponding author upon reasonable request.

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

## Acknowledgements

We thank Joseph Paton, Marta Moita, Alfonso Renart, Tim Hanks, Chris Summerfield and the Mainen, and Pouget Laboratories for helpful discussions on the work presented here. In particular, we would like to thank Jeff Beck and Ingmar Kanitscheider on feedback regarding model conceptualization and implementation. This work was supported by grants from the Champalimaud Foundation (M.I.V., A.G.M., E.E.J.D., Z.F.M.), European Research Council (Advanced Investigator Grants 250334 and 671251, Z.F.M.), Fundação para a Ciência e a Tecnologia (SFRH/BD/33938/2009 to A.G.M., SFRH/BD/ 33274/2008 to M.I.V.), Human Frontier Science Program (Grant RGP0027/2010, Z.F.M. & A.P.), Simons Foundation (Grant 325057, Z.F.M. & A.P.), the University of Geneva (A. P.), the Swiss National Science Foundation (31003A_143707 and 31003A_165831, A.P.), the James S. McDonnell Foundation (Scholar award in Understanding Human Cognition, grant 220020462, J.D.) and the National Institute of Mental Health (R01MH115554, J.D.).

## Author contributions

A.G.M., M.I.V. and Z.F.M. designed the experiments. A.G.M., J.D., Z.F.M. and A.P. designed the models. M.I.V. conducted the experiments with assistance from A.G.M.. M.I.V., A.G.M., J.D. and E.E.J.D. analyzed the data. A.G.M. and J.D. implemented the models. A.G.M., J.D., M.I.V., A.P. and Z.F.M. wrote the manuscript.

## Competing interests

The authors declare no competing interests.
