## [Peer Review File · Nature Communications]

Reviewers' comments:

Reviewer #1 (Remarks to the Author):

Mendonca AG et al., The impact of learning on perceptual decisions and its implication for speed-accuracy trade-offs

The authors describe a study in which rats carried out odor identification and categorization tasks, under varying information levels. Accuracy and reaction time performance in the tasks was quantified using a series of models. First, it was found that a standard DDM with collapsing bounds could not predict RT and accuracy across tasks with a single set of parameters. The authors then developed a model that included a term that accounted for spatial/odor preferences, across trials, in the animal's choice behavior. They found that this model could account well for both accuracy and reaction times, across all conditions. The further compared this model to a series of simpler models and found that the simpler models were not able to account well for the behavior.

This paper addresses an important question about the sources of variability in odor discrimination tasks in the rat. This is an important paradigm used by a large number of labs, and having detailed models of the behavior is critical to understanding the neural systems that give rise to this behavior. The experiments were carefully carried out, the paper is clearly written, and the modeling is developed in detail. I have a few comments which if addressed could increase the clarity of the outcome.

Comments

1. My main comment is that the results are written as if the effect is a change in the perceptual decision boundary. However, the odor identify is confounded with the motor response in this paradigm. This does not really affect any of the results or the modeling. And the interpretation as presented could be correct. However, I think another possible interpretation is that the animals are integrating both a location preference that tracks rewarded locations, and information about the odor. Rodents in particular are strongly spatial, and learn spatial preferences very effectively. So I think the discussion should mention the fact that, the animals may not be shifting their perceptual boundary. Rather they are integrating perception and the reinforced spatial location of choices.

2. My second comment has to do with the assumption that the mean of the perceptual stimulus is given by the exponential function in equation 1. What is the means are fit to the data? What do they look like? Is it possible that a more complex equation 1 could give rise to some of the effects seen in the data? This equation seems like a strong assumption, and it is only backed up by 1 reference. Particularly for mixed odors there might be interactions between the odors?

3. Fig. 6 was unfortunately not very high resolution. It was a bit hard to see the details. Also, in the results it says there should be some red arrows, but my figure is grey scale.

4. The methods that describe the training sequence were a bit unclear. The methods state that, "During training, in phases V-VII, we used..." Was VII testing, as in data collection for the paper, or was this truly training, and all testing followed? Also, the names of the tasks are not completely consistent throughout the paper. It would be useful if one name was adopted for each of the 3 tasks and it was used consistently.

Typos:

"and minimizing least the least square"

The reference, "using the method described in Simpson, Turner, ..." should be just the numbered reference.

"these two steps on hand of making..."

Equation 31 seems to have an error. Or maybe the w is subscript and shouldn't be?

Bruno Averbeck

Reviewer #2 (Remarks to the Author):

This is a well-written paper on a topic of general interest, with solid experimental data used to test/validate distinct models of decision making. The model comparisons are a strength of the paper and the findings are beyond what could be anticipated without such modeling work. It is particularly valuable that the authors were able to account for secondary features of the data after fitting a model to the primary features.

It would be good to see more discussion about the similarity/difference in the dot motion task, where the number of dots is constant and total dot-wise motion is conserved, but proportion that move in one direction over the other varies. In some ways, this is superficially similar to the category task (total number of inputs conserved) but behaves more like the identification task (That is, comment on why lots of dots moving randomly is like a minimal odor signal rather than a 50% mixture of strong, opposing signals/stimuli).

p. 17 "learns less when it is less confident"

I think it difficult to claim this is intuitive given the nature of reward prediction errors and also seems counter to the point made on page 33 in the Methods about the modulation of alphas. Certainly, for a correct response (and the analyzed data in this paper are only based on correct responses in prior trials) reward prediction errors and hence learning would be greater when there is more uncertainty (i.e. less confidence) in the outcome prior to reward delivery. Some further clarification is needed here.

Finally, I think it is worth commenting on whether a 2-dimensional model of decision-making with fixed bounds on the individual variables (such as the leaky-competing accumulator model, LCA, or any simplification of a neural circuit with two competing groups of neurons) could account for the main differences in outcomes across the two sets of stimuli in the task. That is, the reduced 1D model (DDM) is essentially measuring the "difference" in the accumulated evidence associated with each of the two stimuli. However, models such as the LCA, where a bound can be reached when an individual variable accumulating evidence for one of the stimuli hits threshold, are likely to show an inability to reduce response times when the evidence (stimulus-difference) is near zero if each stimulus individually remains strong. Since this is the simplest explanation that would first come to the mind of most readers, it should be addressed as fully as possible.

Fig. 8 is pretty hard to follow. I think it would be helpful if (as I understand it) μ_1 and μ_2 were defined as estimated drift for the correct/stronger stimulus and estimated drift for the incorrect/weaker stimulus, *irrespective of absolute identity*. The connection made to Fig.2A where the axes are Odor A and Odor B is a bit confusing unless we are told in Fig 8(c-f) that we are only looking at trials where Odor A is the correct choice. Also, use of fluctuating "bound" as the titles in this figure when it is the weights that fluctuate and hence the corresponding integration rates that fluctuate is also confusing, since "bound" is so suggestive of the "decision boundary" which could fluctuate but does not do so in this model. If another term other than "bound" can be used please do so, or at least alter to "category boundary".

Minor/typos:

First mention of "DDMs" in the intro (p.2 l.4) would benefit from the words "drift-diffusion models" used later in the text.

p.3 should be "mechanisms are specific" (remove "that")

p.3 "boundary" when talking about the classification boundary: would be easier to follow if a different term is used, given the "bounds" on decision making models. Or at least (like later usage) use "category boundary" whenever using this term.

p.4 should be "situations in which those conditions do not hold"

p.7 On optimality of DDMs: given this paper addresses internal noise it might be worth mentioning that the optimality of DDM assumes noise is external and DDMs are often not optimal when there is internal noise that accumulates (see e.g. Miller & Katz 2013).

p.9 final line I think "effect of" as the mixture contrast is always modulated (i.e. changed) in the expts.

p.12 final line. I think "fit well" is a bit of an overstatement, especially for the right-hand panel showing Response Times of the intermediate difficulties (one curve only passes through 1 of 4 error bars, which already allow for a lot of slack). I think "fit qualitatively" is more appropriate – and is indeed fine for data points not explicitly included in the fitting procedure.

p.14 should be "neither data nor" (not "no")

p.15 should be "multiple terms whose" (not "who")

p. 26 Eq. 10 has a typo

p.31 Please define "lapse bias" in terms of the symbol " I_b " as I assume they are not the same.

p.33 Eq 31 seems wrong

p.39 "foxed" should be "fixed"

Reviewer #3 (Remarks to the Author):

The present manuscript details an accumulator model with a Bayesian front end to adjust trial-by-trial decision weights according to the stimulus/feedback history. I think this is important work, and the field is already moving in this direction to build better models of trial-to-trial effects. It is good to see that this pattern of behavior occurs in rats, where hopefully we can soon understand the neural circuits that relate to these sequential adjustments.

That said, the message of the article can be expressed as "a front end that is sensitive to experimental information is better than one that is not". Basically, the authors embed a learning rule on decision weights that are then used in the stochastic integration process of a Wiener diffusion model. The learning rule is "Bayesian" in the sense that it updates expectations in terms of reward (and there is some leakiness to this process). They then show that this model does better than models that (1) are

standard DDM applications with only variance in the momentary integration, (2) assume random weight sampling from trial to trial, (3) assume Bayesian updating of bias terms, or (4) an RL-DDM structure with point estimates not full distributions over weights. Honestly, I didn't find the set of comparison models to be particularly exciting. For (1), I don't think anyone in this field would seriously use a model of choice RT that didn't assume some form of between-trial variability. So, the conclusion that the "DDM" doesn't capture the pattern of data can be ignored, in my opinion. The full DDM that they cite actually has three sources of between-trial variability, which have been shown to be important in capturing the typical SAT pattern. Moving on to (2), the authors incorporate some component of trial-to-trial variability in the base model, and conclude that it can capture the key effects in the choice RT data. However, it does not capture the sequential dependencies. The author acknowledges that this shouldn't happen as the model is insensitive to the task and response information, unlike the Bayes-DDM. So, again, we can probably rule this out as a fair comparison given that the metric for success has moved from capturing the data to capturing sequential dependencies (which is unfair to their compared DDM). (3) gets more interesting, although it's really difficult for me to tell that bias alone is causing the differences that are clearly observed in the RT data across the two tasks. It seems much more likely that drift should change, or potentially both, so it is good that (3) confirms this. Finally, (4) is interesting as it shows that the full distribution of weight uncertainty is important in the trial-to-trial adjustments. This was sort of anticipated (by me at least) as those initial trials would suffer from having strong adjustments in response to very little data. The uncertainty in the weights is probably what drives this effect (although I don't see an analysis of trial-level fit, which might help to unravel the contribution of the full weights rather than the point estimates).

There are several good models already that included trial-to-trial dependencies, such as adjusting the threshold, drift rate, or drift criterion. For example, the RL-DDM work that Michael Frank has done really helps to guide the drift rates through time (and some starting points with neural covariates). There are also several papers on sequential effects in choice RT data (only listing a few of these below). Perhaps more importantly is that there are already Bayesian-like models of category representations that evolve with experience (stimulus and feedback). These models show the benefits of trial-to-trial adjustments in the representations of the stimuli, rather than simply adjusting decision weights. For example, a new paper by Brandon Turner explicitly compares the types of evolving representations in dynamic environments. So I'm not sure if there is a harsh citation limit in this format or not, but it would be good to acknowledge this other work or use their models as the starting point for comparing the contribution of the Bayes-DDM.

Finally, I couldn't really figure out how the models were being fit to data. They were apparently simulated for 100,000 trials, but how did this match up to the data? They are using some Monte Carlo method, but it would be good to know more about this without tracking down the other paper. It would also be good to fit the model to the full RT distribution rather than just the mean, but this is challenging (but not impossible! There are methods out there for this).

RL-DDM

http://ski.clps.brown.edu/papers/CollinsEtAl_RLWMPST.pdf

http://ski.clps.brown.edu/papers/FrankEtAl_RLDDM_fMRI EEG.pdf

http://ski.clps.brown.edu/papers/CockburnEtAl_ChoiceBias.pdf

category learning

<http://catlab.psy.vanderbilt.edu/wp-content/uploads/NP-PsyRev1997.pdf>

<https://turnermbcn.files.wordpress.com/2017/08/turnervanzandtbrown2011.pdf>

<https://psycnet.apa.org/record/2019-20305-001>

absolute identification

<https://psycnet.apa.org/record/2008-04236-005>

sequential effects:

<https://link.springer.com/article/10.1007/s11336-010-9172-6>

<https://amstat.tandfonline.com/doi/abs/10.1080/01621459.2016.1194844#.XLdg95NKjBI>

Reviewers' comments:

Reviewer #1 (Remarks to the Author):

Mendonca AG et al., The impact of learning on perceptual decisions and its implication for speed-accuracy trade-offs

The authors describe a study in which rats carried out odor identification and categorization tasks, under varying information levels. Accuracy and reaction time performance in the tasks was quantified using a series of models. First, it was found that a standard DDM with collapsing bounds could not predict RT and accuracy across tasks with a single set of parameters. The authors then developed a model that included a term that accounted for spatial/odor preferences, across trials, in the animal's choice behavior. They found that this model could account well for both accuracy and reaction times, across all conditions. The further compared this model to a series of simpler models and found that the simpler models were not able to account well for the behavior.

This paper addresses an important question about the sources of variability in odor discrimination tasks in the rat. This is an important paradigm used by a large number of labs, and having detailed models of the behavior is critical to understanding the neural systems that give rise to this behavior. The experiments were carefully carried out, the paper is clearly written, and the modeling is developed in detail. I have a few comments which if addressed could increase the clarity of the outcome.

We thank the reviewer for the positive words and hope to have addressed the comments below and in our revised manuscript.

Comments

1. My main comment is that the results are written as if the effect is a change in the perceptual decision boundary. However, the odor identify is confounded with the motor response in this paradigm. This does not really affect any of the results or the modeling. And the interpretation as presented could be correct. However, I think another possible interpretation is that the animals are integrating both a location preference that tracks rewarded locations, and information about the odor. Rodents in

particular are strongly spatial, and learn spatial preferences very effectively. So I think the discussion should mention the fact that, the animals may not be shifting their perceptual boundary. Rather they are integrating perception and the reinforced spatial location of choices.

The reviewer is correct that there is a location preference that tracks rewarded locations. This is in fact reflected in our modeling by having a bias term that tracks recent rewards regardless of stimulus quality. However, the effect of this bias is insufficient to explain the results seen in categorization task for both changes in RTs (the model is too slow) and choice bias, as they are not modulated by stimulus quality (**Suppl. Fig. 15**). Adding random weight fluctuations helps explain categorization performance and RTs (**Suppl. Fig. 18**) but doesn't explain the changes seen in choice bias for each stimulus (**Suppl. Fig. 18g**). Thus the need to combine both reinforced spatial location (through bias) and category boundary fluctuations through learning (**Fig. 5**). We have made this point clearer in the discussion, pg. 21 and 22:

*In both tasks, rats feature a location preference that tracks rewarded locations (**Fig. 6**). However, this location preference does not appear to simply follow a general reward bias, as such a reward bias is insufficient to explain the behavior seen in categorization task (**Suppl. Fig. 15**). Adding random weight fluctuations helped explain categorization performance and RTs (**Suppl. Fig. 18**) but did not match the stimulus-dependent choice bias pattern (**Suppl. Fig. 18g**). We could explain the whole set of behaviors by combining both reinforced spatial location (through bias) and category boundary fluctuations through learning (**Fig. 5**).*

2. My second comment has to do with the assumption that the mean of the perceptual stimulus is given by the exponential function in equation 1. What are the means are fit to the data? What do they look like? Is it possible that a more complex equation 1 could give rise to some of the effects seen in the data? This equation seems like a strong assumption, and it is only backed up by 1 reference. Particularly for mixed odors there might be interactions between the odors?

One possible way of fitting the data (and what we believe the reviewer is alluding to here) is to consider each drift rate separately and fit the mean performance and reaction time of each stimulus. However, in this case, the number of free parameters increases dramatically, as each stimulus would then be attributed a rate. This, in our view, brings two problems: one is related to computational power, as the increased number of parameters would make the convergence to a solution even slower. This will be even more troublesome when considering the conditional effects. Clearly drift rates (i.e., the process within the integration) alone cannot explain these. Secondly, by adding more parameters, we fall the risk of overfitting the data, as we would be adding 6 additional parameters per task.

In that light, our decision to have a power law relationship was based on various factors. The first one is a purely behavioral one and has to do with the fact that our stimulus design suggests a logarithmic relationship between stimulus and response, as each step of concentration dilution is a logarithmic one. One suggestion could be to use a purely logarithmic function, but this has a problematic issue in our view: what to do when the input is zero? A second factor we took into account is the usefulness of power laws in exploring logarithmic relationships, without the constraint of non-zero concentration. Power laws allow for the linearization of the relation between drift and concentration in log-log space. Thirdly, power laws are scale-invariant, that is, that by scaling in a log-step we allow proportional scaling of drift rate, which when considering the interleaved scaling seems to make perfect sense (**Fig. 3**). Lastly, power law scaling is the bread and butter of DDM models seen in the literature and in a multitude of different behavioral paradigms (Ratcliff & Rouder, 1998; Palmer, Huk & Shadlen, 2005; Ditterich, 2006; Ratcliff & McKoon, 2008; Brunton, Botvinick & Brody, 2013). We added the following clarification to the main text regarding our usage of a power law relationship (pg. 7):

We implemented a power law relationship between stimulus and drift for two reasons: one is purely behavioral as our stimulus design suggests a logarithmic relationship between stimulus and response, with each step of concentration dilution being a logarithmic one; two is the usefulness of power laws in exploring logarithmic relationships, without the constraint of non-zero concentration. Power laws allow for the linearization of the relation between drift and concentration in a log-log space.

Considering the case of interaction between odors, we know from past work that, in the categorization task, odor mixtures are classified predictably and simply by the ratio of these two odorants, invariant of intensity (Uchida & Mainen, 2008). As discussed in that work, this rules out at least gross interactions between the chosen odorants.

3. Fig. 6 was unfortunately not very high resolution. It was a bit hard to see the details. Also, in the results it says there should be some red arrows, but my figure is grey scale.

We apologize for the low resolution and hope that the figures are clearer in the revised manuscript. Additionally, the text should have read "black arrows" and not "red arrows" - a typo that we have now corrected.

4. The methods that describe the training sequence were a bit unclear. The methods state that, "During training, in phases V-VII, we used..." Was VII testing, as in data collection for the paper, or was this truly training, and all testing followed? Also, the names of the tasks are not completely consistent throughout the paper. It would be useful if one name was adopted for each of the 3 tasks and it was used consistently.

This was a typo from our part and for that, we apologize. Training phases in which we used adaptive algorithms were V and VI. Testing for each task was done immediately after training. We have corrected this in Methods so to avoid ambiguity.

We thank the reviewer for pointing out some inconsistent naming throughout the manuscript. We have reviewed and made the naming consistent.

Typos:

"and minimizing least the least square"

The reference, "using the method described in Simpson, Turner, ..." should be just the numbered reference.

"these two steps on hand of making..."

Equation 31 seems to have an error. Or maybe the w is subscript and shouldn't be?

Thank you for pointing out these typos. We have corrected them.

Reviewer #2 (Remarks to the Author):

This is a well-written paper on a topic of general interest, with solid experimental data used to test/validate distinct models of decision making. The model comparisons are a strength of the paper and the findings are beyond what could be anticipated without such modeling work. It is particularly valuable that the authors were able to account for secondary features of the data after fitting a model to the primary features.

We appreciate the reviewer's remarks. We agree that one of the strongest suits of our work is the ability to account for secondary features of the data while focusing on primary features.

It would be good to see more discussion about the similarity/difference in the dot motion task, where the number of dots is constant and total dot-wise motion is conserved, but proportion that move in one direction over the other varies. In some ways, this is superficially similar to the category task (total number of inputs conserved) but behaves more like the identification task (That is, comment on why lots of dots moving randomly is like a minimal odor signal rather than a 50% mixture of strong, opposing signals/stimuli).

We agree with the reviewer's assessment that intuitively the dot motion task (RDM) seems to share some properties with the categorization task due to, as stated, "total number of inputs conserved". However, we would like to point out that in the RDM the stimulus is a mixture of $(1-coh)\%$ randomly moving dots, and $coh\%$ coherently moving dots, but only the latter is informative about the choice. The odor categorization task also has two components, but both are "informative" (i.e., have an associated choice), and the question is which one is stronger. In the odor identification task, only a single component is presented (+ background noise). We believe that in that sense, the identification task is more

similar to RDM, as the reviewer points out. We thank the reviewer for pointing out this interesting question and added the following paragraph to the manuscript, pg. 21:

The weak RT modulation in the categorization task raises the interesting question regarding its relation to the random dot motion (RDM) discrimination task frequently used in primates^{1,2,49,50}. The degree at which they differ has been a long-time debate in the field of olfactory discrimination^{15-17,51}. More specifically, the RDM task requires identifying the coherent motion of a subset of coherently moving dots that are masked by otherwise randomly moving dots, such that only those coherently moving dots are informative about the correct choice. The odor categorization task also has two stimulus components, but they are both informative, as the decision-maker needs to compare their strength, which makes it conceptually different from RMD.

p. 17 “learns less when it is less confident”

I think it difficult to claim this is intuitive given the nature of reward prediction errors and also seems counter to the point made on page 33 in the Methods about the modulation of alphas. Certainly, for a correct response (and the analyzed data in this paper are only based on correct responses in prior trials) reward prediction errors and hence learning would be greater when there is more uncertainty (i.e. less confidence) in the outcome prior to reward delivery. Some further clarification is needed here.

The reviewer is correct in pointing out this oversimplification from our part. What we meant was that when an error trial occurs for long trials the subject is less confident and thus learns less than when fast and confident. But this only occurs for incorrect trials, meaning that what we depicted here is the incomplete picture. We have re-written the paragraph and hope it is clearer now, pg. 17 and 18:

Therefore, for incorrect trials, the error term in this learning rule is decreasing as confidence decreases over time, which is to say the model learns less when it is less confident. Interestingly, for correct trials, the relationship is inverted, as the model learns more

strongly when less confident, which makes intuitive sense: in confident correct trials there is no more information to be gained.

Finally, I think it is worth commenting on whether a 2-dimensional model of decision-making with fixed bounds on the individual variables (such as the leaky-competing accumulator model, LCA, or any simplification of a neural circuit with two competing groups of neurons) could account for the main differences in outcomes across the two sets of stimuli in the task. That is, the reduced 1D model (DDM) is essentially measuring the “difference” in the accumulated evidence associated with each of the two stimuli. However, models such as the LCA, where a bound can be reached when an individual variable accumulating evidence for one of the stimuli hits threshold, are likely to show an inability to reduce response times when the evidence (stimulus-difference) is near zero if each stimulus individually remains strong. Since this is the simplest explanation that would first come to the mind of most readers, it should be addressed as fully as possible.

We thank the reviewer for pointing out this possibility. To test it we have implemented two LCA variants (following Usher & McClelland, 2001): (i) time-invariant boundaries (8 parameters with lapses, etc.), and (ii) linearly collapsing boundaries (9 parameters). The model doesn't update the input weights across trials, but its parameters are tuned to best match overall psychometric and chronometric curves across both conditions. Both variants are able to fit both conditions with the same set of parameters (**Supplementary Figs. 7,8**). However, they fail to capture the trial-by-trial changes in choice bias (**Supplementary Figs. 7d,7g,8d,8g**).

This indicates that some form of learning is essential to capture this property of our data. We could have derived learning rules for LCA models, but decided against it, as - for our task - the optimal decision model was the DDM. Nonetheless, we believe that adding some sort of reinforcement learning rule with LCA (as the kind done in RL-DDM) might be able to replicate most of our data. However, we hope that the reviewer agrees that, due to the already large number of compared model variants, adding yet another heuristic model is beyond the scope of this study. We have added these models to our supplementary data (**Supplementary Figs. 4,7,8**).

Fig. 8 is pretty hard to follow. I think it would be helpful if (as I understand it) μ_1 and μ_2 were defined as estimated drift for the correct/stronger stimulus and estimated drift for the incorrect/weaker stimulus, *irrespective of absolute identity*. The connection made to Fig.2A where the axes are Odor A and Odor B is a bit confusing unless we are told in Fig 8(c-f) that we are only looking at trials where Odor A is the correct choice.

Fig. 8 are in fact trials that Odor A was the correct choice. We agree that this analysis is a bit hard to follow, thus, for clarity, re-wrote both the main text and figure captions. Hopefully, these corrections will make the figures easier to understand.

Also, use of fluctuating "bound" as the titles in this figure when it is the weights that fluctuate and hence the corresponding integration rates that fluctuate is also confusing, since "bound" is so suggestive of the "decision boundary" which could fluctuate but does not do so in this model. If another term other than "bound" can be used please do so, or at least alter to "category boundary".

We agree with the author that the usage of the word bound might mislead the readers and confuse them as to whether we refer to the "decision boundary" or the "category boundary". We reviewed and reinforced the correct naming to avoid further confusion.

Minor/typos:

First mention of "DDMs" in the intro (p.2 l.4) would benefit from the words "drift-diffusion models" used later in the text.

p.3 should be "mechanisms are specific" (remove "that")

p.3 "boundary" when talking about the classification boundary: would be easier to follow if a different term is used, given the "bounds" on decision making models. Or at least (like later usage) use "category boundary" whenever using this term.

p.4 should be "situations in which those conditions do not hold"

Thank you for these suggestions. We have implemented them in the revised manuscript.

p.7 On optimality of DDMs: given this paper addresses internal noise it might be worth mentioning that the optimality of DDM assumes noise is external and DDMs are often not optimal when there is internal noise that accumulates (see e.g. Miller & Katz 2013).

We thank the reviewer for the suggested reference, as it is indeed related to our work, as we consider how variability impacts decision-making performance. The biggest difference between our work and that of Miller & Katz (2013) is the time-scale of this variability. In Miller & Katz (2013), variability emerges at the time-scale of individual decisions, making the DDM a suboptimal decision-making strategy. In our case, we do not assume internal variability at such time-scales, such that the DDM remains optimal (for a more detailed mathematical description, see Drugowitsch, Mendonça, Mainen & Pouget, 2019). Our variability instead emerges between individual decisions through a change in the category boundary. Such a change is rational if one assumes that the environment slowly changes, as the animals might. It only becomes sub-optimal in stationary environments, as that of our experiments. Therefore, while the decision-maker might be rational, they might not be optimal. It would be optimal to keep a fixed category boundary, but this would not result in the sequential choice dependencies observed in animal behavior.

Given the different time-scales of internal variability, we hope that the reviewer agrees with not adding the suggested reference, in order to not side-track the reader

p.9 final line I think "effect of" as the mixture contrast is always modulated (i.e. changed) in the expts.

Thank you for pointing this out. We have corrected the wording.

p.12 final line. I think "fit well" is a bit of an overstatement, especially for the right-hand panel showing Response Times of the intermediate difficulties (one curve only passes through 1 of 4

error bars, which already allow for a lot of slack). I think “fit qualitatively” is more appropriate – and is indeed fine for data points not explicitly included in the fitting procedure.

We have reworded the sentence in line with the reviewer’s suggestion.

p.14 should be “neither data nor” (not “no”)

p.15 should be “multiple terms whose” (not “who”)

Both have been corrected.

p. 26 Eq. 10 has a typo

The equation is now corrected.

p.31 Please define “lapse bias” in terms of the symbol “ l_b ” as I assume they are not the same.

We thank the reviewer for pointing this inconsistency. We have renamed the parameters in both the main text and Supplementary material.

p.33 Eq 31 seems wrong

We have corrected Equation 31.

p.39 “foxed” should be “fixed”

Corrected.

Reviewer #3 (Remarks to the Author):

The present manuscript details an accumulator model with a Bayesian front end to adjust trial-by-trial decision weights according to the stimulus/feedback history. I think this is important work, and the field is already moving in this direction to

build better models of trial-to-trial effects. It is good to see that this pattern of behavior occurs in rats, where hopefully we can soon understand the neural circuits that relate to these sequential adjustments.

We appreciate the words and positive reaction from the reviewer.

That said, the message of the article can be expressed as “a front end that is sensitive to experimental information is better than one that is not”. Basically, the authors embed a learning rule on decision weights that are then used in the stochastic integration process of a Wiener diffusion model. The learning rule is “Bayesian” in the sense that it updates expectations in terms of reward (and there is some leakiness to this process). They then show that this model does better than models that (1) are standard DDM applications with only variance in the momentary integration, (2) assume random weight sampling from trial to trial, (3) assume Bayesian updating of bias terms, or (4) an RL-DDM structure with point estimates not full distributions over weights. Honestly, I didn’t find the set of comparison models to be particularly exciting. For (1), I don’t think anyone in this field would seriously use a model of choice RT that didn’t assume some form of between-trial variability. So, the conclusion that the “DDM” doesn’t capture the pattern of data can be ignored, in my opinion. The full DDM that they cite actually has three sources of between-trial variability, which have been shown to be important in capturing the typical SAT pattern.

Moving on to (2), the authors incorporate some component of trial-to-trial variability in the base model, and conclude that it can capture the key effects in the choice RT data. However, it does not capture the sequential dependencies. The author acknowledge that this shouldn’t happen as the model is insensitive to the task and response information, unlike the Bayes-DDM. So, again, we can probably rule this out as a fair comparison given that the metric for success has moved from capturing the data to capturing sequential dependencies (which is unfair to their compared DDM).

Regarding (1), we agree that a large body of work - in particular in human decision-making (e.g., the work of Ratcliff & colleagues) - uses DDM variants with different forms of between-trial variability. These are commonly motivated by empirical observations, such as slower error

responses, rather than normative considerations. The (by now sizable) animal literature (e.g., the work of Shadlen & colleagues; also Palmer, Huk & Shadlen for human psychophysics), in contrast, routinely uses DDM variants without such variability, based on normative considerations and neurophysiological observations. Therefore, in context of the latter, it is indeed not a given that DDMs without such non-normative variabilities are unable to capture the observed behavior. Therefore, we do not consider vanilla DDMs as a “strawman”, as they have been seriously and successfully used in previous work, and so also deserve serious consideration in our work.

Regarding (2), we would like to emphasize that it is suboptimal in our task to feature sequential choice dependencies. This is because the stimulus is drawn randomly and independently across consecutive trials, such that the stimulus and choice in the previous trial is completely uninformative about the stimulus and choice in the current trial. Therefore, a model that is “sensitive” to the task shouldn’t feature sequential choice dependencies. Only once we introduce the *wrong* assumption that the environment slowly changes across time do we recover these sequential choice dependencies. Thus, as for (1), a model that does not feature sequential choice dependencies is by no means a “strawman”, and requires serious consideration, as we do in our work. Nonetheless, we would like to emphasize that it wasn’t our goal to make an unfair comparison between a more structured model against a simpler yet useful model such as the DDM. On the contrary, we aimed to show that our apparent extra “source of noise” is actually quite well explained by the assumption that the world is ever-changing. We believe that this additional step does not reject, but in fact strengthens the role of DDM in discriminating stimuli.

(3) gets more interesting, although it’s really difficult for me to tell that bias alone is causing the differences that are clearly observed in the RT data across the two tasks. It seems much more likely that drift should change, or potentially both, so it is good that (3) confirms this.

It is important to note that changes in bias alone do not cause by itself the observed differences in RT data. There is a need to add some form of variability in drift, in the form of random updating (**Suppl. Fig. 18**) or structured learning (**Fig. 5**). This is in agreement with the point brought forward by reviewer #1. We made this point clearer in the discussion.

Finally, (4) is interesting as it shows that the full distribution of weight uncertainty is important in the trial-to-trial adjustments. This was sort of anticipated (by me at least) as those initial trials would suffer from having strong adjustments in response to very little data. The uncertainty in the weights is probably what drive this effect (although I don't see an analysis of trial-level fit, which might help to unravel the contribution of the full weights rather than the point estimates).

We are not completely sure what effects that are explained by weight uncertainty the reviewer is referring to. In fact, it is unlikely that changes in weight uncertainty play a major role in the pattern of sequential dependencies we observe, for multiple reasons. First, weight uncertainty in Bayes-DDM decreases with every received feedback (i.e., rewarded vs. non-rewarded choices), and subsequently increases again as the decision-maker believes that these weights change across consecutive trials. Thus, the weight uncertainty will quickly reach a steady-state around which it hovers across consecutive trials. Second, we show that RL-DDM models the data as well as Bayes-DDM, even though RL-DDM only tracks weight point estimates. Both make weight uncertainty an unlikely contributor to explaining the sequential choice dependency pattern we observe. Instead, as we point out in the main text, it appears that decision confidence modulates the strength of the weight change (i.e., the magnitude, not necessarily its impact on the uncertainty), which results in this pattern.

There are several good models already that included trial-to-trial dependencies, such as adjusting the threshold, drift rate, or drift criterion. For example, the RL-DDM work that Michael Frank has done really helps to guide the drift rates through time (and some starting points with neural covariates). There are also several papers on sequential effects in choice RT data (only listing a few of these below). Perhaps more importantly is that there are already Bayesian-like models of category representations that evolve with experience (stimulus and feedback). These models show the benefits of trial-to-trial adjustments in the representations of the stimuli, rather than simply adjusting decision weights. For example, a new paper by Brandon Turner explicitly compares the types of evolving representations in dynamic environments. So I'm not sure if there is a harsh citation

limit in this format or not, but it would be good to acknowledge this other work or use their models as the starting point for comparing the contribution of the Bayes-DDM.

We thank the reviewer for bringing these studies to our attention. After some close examination, we have some comments regarding the mentioned models.

The RL-DDM model from Michael Frank does in fact combine reinforcement learning with the DDM model. However, there is a fundamental difference between their model and ours. In Michael Frank's RL-DDM the DDM drives the decision process on top of a learning process that is learning decision values. This is different from our particular case, as the DDM in fact drives learning through the DDM. As an example, when we consider our own version of RL-DDM it is the decaying threshold that drives the prediction error component of RL. This two-lane interaction between RL and DDM is what distinguishes our own version with Michael Frank's model. Additionally, it is important to note that in all cited experiments the subjects are not exposed to noisy ambiguous stimuli, but to environments in which subjects are still learning the true contingencies between actions and decision value. Our study is in fact similar but adds one extra layer to this question: the interaction between stimulus discrimination and learning. This is particularly striking when we consider that our dataset is focused on overall stabilized performance and not task learning.

Regarding the 2011 work of Brandon Turner, we do agree that this model is in many ways relevant to our work and related to what we address here. However, it partially relies on signal detection theory, which doesn't allow them to account for response times, whereas response times are essential in our modeling approach. In fact, in Turner, Van Zandt and Brown, page 609, they discuss: "A second major shortcoming of our model, and of the SDT framework in general, is that it fails to make predictions about response times." Our model expands this by incorporating DDM and making predictions on how performance and RT interact.

More recently this model has been upgraded, as the reviewer points out, in a review that compares multiple models (Turner, 2019). We would like to point out that in this comparison, both stimulus and environment are dynamic, which clearly does not apply to our behavioral data. In

particular, the effects of learning are measured by manipulating a dynamic categorization task. Our model does assume, wrongly, a dynamic environment, making this study indeed relevant to our manuscript. However, it is unclear to us how this model would predict dynamic fluctuations of biases if the environment is not manipulated further. We believe our model expands on this by showing that even in stable environments one can still see the effects of learning by assuming nonstationarity. However, we do agree it is relevant to our study and have included it in our discussion.

The remaining papers are in fact all relevant to our study, and for that, we appreciate the reviewer bringing them to our attention. However, these models do feel different in nature as they explore learning in dynamic environments (Turner, 2019) contrasting with our fixed odor discrimination environment, effects of long and short term trace memories (Brown et al, 2008; Collins et al, 2017; Kim et al, 2017) contrasting with our fixed equiprobable stimuli, and prediction of performance betterment as learning dissipates (Frank et al, 2015) which we don't see in our data. All these small differences taken together make our model novel as it combines RTs, learning and stimulus categorization.

We have incorporated some of these citations in our manuscripts and added discussion points that we believe are important, pg. 22 and 23:

Previous work has combined learning with stimulus categorization in the field of decision-making, albeit in different contexts such as learning phases⁵², dynamic environments⁵³ and manipulation of short term memory⁵⁴⁻⁵⁶. In particular, in Frank et al. (2015), Frank and colleagues combined RL with DDM models, and called this combination RL-DDM^{52,55,57}, as we do in this work. However, there is a fundamental difference between the two models. In Frank et al.'s RL-DDM, the DDM drives the decision process on top of a reinforcement learning process that is learning decision values. This is different from our case, in which learning is happening within the DDM. Thus, rather than trying to learn the consequences of one's action, our study focuses directly on learning how to perform good actions in the first place. This is particularly striking considering that our task itself doesn't change over time, making continual learning unnecessary. Learning thus only emerges due to the faulty assumption that the task might change. This makes it similar to Turner et al. (2011), who also explored dynamic stimulus learning⁵⁸. However, this study differs from ours as it relies on signal

detection theory, which does not account for response times, whereas RTs are essential in our modelling approach. More recently, Turner extended this model to explore the role of learning in dynamic environments⁵³. Our model expands on this by showing that it is possible to observe effects of learning even in stable environments as long as the decision-maker wrongly assumes the environment to be dynamic.

Finally, I couldn't really figure out how the models were being fit to data. They were apparently simulated for 100,000 trials, but how did this match up to the data? They are using some Monte Carlo method, but it would be good to know more about this without tracking down the other paper. It would also be good to fit the model to the full RT distribution rather than just the mean, but this is challenging (but not impossible! There are methods out there for this).

We haven't been very clear on how we fit the model to the observed behavior. In particular, we didn't sufficiently distinguish which parts of the model fitting description referred to simulated, and which part to observed behavior. We have now added these qualifiers throughout the 'Model fitting' section in methods. In this section, we are now furthermore referring back to how we generate the simulations, which is described in 'Drift-diffusion model for decision-making' in Methods. We hope that this clarifies how we are performing the model fits.

Regarding the full RT distribution, we would like to highlight that we are at the limit of what we can do with this dataset. We not only fit mean and standard deviations of RT distributions for all stimuli (8 per task), but furthermore perform these fits also for conditional distributions (8 stimuli conditioned over $8 = 64$ conditions). In order to perform fits of the whole RT distribution across all conditions, we would require an extremely high number of trials. Additionally, we would like to highlight that our experiments are based on a randomized sequence of trials. Even if we considered the sequence of trials and their influence the RT distribution would be extremely difficult to analyze. This is a known problem when comparing influences of outcomes to RT distribution. For instance, in (Craigmire, Peruggia & Van Zandt, 2010), the authors designed the

experiment (page 615) “so that the stimulus sequences were exactly the same regardless of which task a subject was to perform”. This ad-hoc strategy is extremely useful when one wants to compare effects on the same subjects with different contexts. However, when we designed our experiment we did not consider this design. Thus, we hope that the reviewer understands that data sparsity and computational considerations make it almost impossible for us to attempt to fit full RT distributions, even though we have done so in previous work (e.g., Drugowitsch et al., 2012). Furthermore, we doubt that any of the main points would change, were we to do so.

REFERENCES:

- Brunton, B. W., Botvinick, M. M. & Brody, C. D. Rats and Humans Can Optimally Accumulate Evidence for Decision-Making. *Science* (80-.). 340, 95–98 (2013).
- Brown, S. D., Marley, A. A. J., Donkin, C. & Heathcote, A. An Integrated Model of Choices and Response Times in Absolute Identification. *Psychol. Rev.* 115, 396–425 (2008).
- Craigmile, P. F., Peruggia, M. & Van Zandt, T. Hierarchical Bayes models for response time data. *Psychometrika* 75, 613–632 (2010).
- Cockburn, J., Collins, A. G. E. & Frank, M. J. A Reinforcement Learning Mechanism Responsible for the Valuation of Free Choice. *Neuron* 83, 551–557 (2014).
- Collins, A. G. E., Albrecht, M. A., Waltz, J. A., Gold, J. M. & Frank, M. J. Interactions Among Working Memory, Reinforcement Learning, and Effort in Value-Based Choice: A New Paradigm and Selective Deficits in Schizophrenia. *Biol. Psychiatry* 82, 431–439 (2017).
- Ditterich, J. Stochastic models of decisions about motion direction: Behavior and physiology. *Neural Networks* 19, 981–1012 (2006).
- Drugowitsch, J., Mendonça, A. G., Mainen, Z. F. & Pouget, A. Learning optimal decisions with confidence. *bioRxiv* 244269 (2019). doi:10.1101/244269
- Drugowitsch, J., Moreno-Bote, R., Churchland, A. K., Shadlen, M. N. & Pouget, A. The Cost of Accumulating Evidence in Perceptual Decision Making. *J. Neurosci.* 32, 3612–3628 (2012).

Frank, M. J. et al. FMRI and EEG predictors of dynamic decision parameters during human reinforcement learning. *J. Neurosci.* 35, 485–494 (2015).

Kim, S., Potter, K., Craigmile, P. F., Peruggia, M. & Van Zandt, T. A Bayesian Race Model for Recognition Memory. *J. Am. Stat. Assoc.* 112, 77–91 (2017).

Miller, P. & Katz, D. B. Accuracy and response-time distributions for decision-making: Linear perfect integrators versus nonlinear attractor-based neural circuits. *J. Comput. Neurosci.* 35, 261–294 (2013).

Palmer, J., Huk, A. C. & Shadlen, M. N. The effect of stimulus strength on the speed and accuracy of a perceptual decision. *J. Vis.* 5, 1–1 (2005).

Palmeri, T. J. & Nosofsky, R. M. An exemplar-based random work model of speeded classification learning. *Psychol. Rev.* 104, 266–300 (1997).

Ratcliff, R. & McKoon, G. The Diffusion Decision Model: Theory and Data for Two-Choice Decision Tasks. *Neural computation* 20, 873–922 (2008).

Ratcliff, R. & Rouder, J. N. Modeling Response Times for Two-Choice Decisions. *Psychol. Sci.* 9, 347–356 (1998).

Turner, B. M. Toward a Common Representational Framework for Adaptation. *Psychol. Rev.* (2019). doi:10.1037/rev0000148

Turner, B. M., Van Zandt, T. & Brown, S. A Dynamic Stimulus-Driven Model of Signal Detection. *Psychol. Rev.* 118, 583–613 (2011).

Uchida, N. & Mainen, Z. F. Odor concentration invariance by chemical ratio coding. *Front. Syst. Neurosci.* 1, 1–6 (2008).

Usher, M. & McClelland, J. L. The time course of perceptual choice: The leaky, competing accumulator model. *Psychol. Rev.* 108, 550–592 (2001).

REVIEWERS' COMMENTS:

Reviewer #1 (Remarks to the Author):

The authors have addressed my concerns. I have no further comments.

Bruno Averbeck

Reviewer #2 (Remarks to the Author):

All my previous positive statements remain, while my prior criticisms have been addressed well in this updated version.

Reviewer #3 (Remarks to the Author):

I'm happy with the revisions the authors have provided. The main issues I had with the previous round was the connection to other DDM-category learning RL models and the details of the methods. Both of these have been addressed well. I think this makes a very nice contribution to the literature.